# Erythropoietin directly remodels the clonal composition of murine hematopoietic multipotent progenitor cells

Almut S Eisele[1], Jason Cosgrove[1], Aurelie Magniez[1], Emilie Tubeuf[1], Sabrina Tenreira Bento[1], Cecile Conrad[1], Fanny Cayrac[1], Tamar Tak[1], Anne-Marie Lyne[1], Jos Urbanus[2], Leïla Perié[1]*

[1]Institut Curie, Université PSL, Sorbonne Université, CNRS UMR168, Laboratoire Physico Chimie Curie, Paris, France; [2]Netherlands Cancer Institute, Amsterdam, Netherlands

**Abstract** The cytokine erythropoietin (EPO) is a potent inducer of erythrocyte development and one of the most prescribed biopharmaceuticals. The action of EPO on erythroid progenitor cells is well established, but its direct action on hematopoietic stem and progenitor cells (HSPCs) is still debated. Here, using cellular barcoding, we traced the differentiation of hundreds of single murine HSPCs, after ex vivo EPO exposure and transplantation, in five different hematopoietic cell lineages, and observed the transient occurrence of high-output myeloid-erythroid-megakaryocyte-biased and myeloid-B-cell-dendritic cell-biased clones. Single-cell RNA sequencing analysis of ex vivo EPO-exposed HSPCs revealed that EPO induced the upregulation of erythroid associated genes in a subset of HSPCs, overlapping with multipotent progenitor (MPP) 1 and MPP2. Transplantation of barcoded EPO-exposed MPP2 confirmed their enrichment in myeloid-erythroid-biased clones. Collectively, our data show that EPO does act directly on MPP independent of the niche and modulates fate by remodeling the clonal composition of the MPP pool.

*For correspondence:
leila.perie@curie.fr

**Competing interest:** The authors declare that no competing interests exist.

## Editor's evaluation

This paper will be of broad interest to readers in the field of cytokine signaling, experimental hematology, and clinical hematology. Erythropoietin is one of the most widely used cytokines clinically, but the cells it exerts its effects on has been debated. This study has combined clonal lineage tracing and single-cell sequencing to understand the cell population that responds to erythropoietin and indicates that erythropoietin acts directly on multipotent progenitors to transiently modulate their output.

## Introduction

Erythrocytes are the most numerous hematopoietic cells in our body and are constantly renewed (*Sender et al., 2016*). The major inducer of erythroid cell development in steady state and anemic conditions is the cytokine erythropoietin (EPO) (*Richmond et al., 2005*). Recombinant EPO is widely used to treat anemia and is one of the most sold biopharmaceuticals (*Walsh, 2014*). Previously, EPO was thought to solely target erythroid-committed progenitors and induce their increased proliferation and survival via the EPO receptor (EPOR) (*Koury, 2016*). Recently, EPO has also been suggested to act on hematopoietic stem and progenitor cells (HSPCs) (*Cheshier et al., 2007*; *Shiozawa et al.,*

2010; *Grover et al., 2014*; *Giladi et al., 2018*; *Yang et al., 2017*; *Tusi et al., 2018*; *Singh et al., 2018*; *Dubart et al., 1994*), but the nature of EPO's effect on HSPC fate remains unresolved despite potential adverse side effects during long-term EPO usage in the clinics and associations of high EPO levels with leukemias (*Ma et al., 2010*; *Weinreb et al., 2020*; *Li et al., 2015*).

It is well established that EPO can induce HSPCs to cycle, as evidenced by a number of bulk and single-cell studies in vitro and in vivo (*Cheshier et al., 2007*; *Giladi et al., 2018*; *Yang et al., 2017*; *Singh et al., 2018*; *Dubart et al., 1994*). Its role in modulating HSPC fate is less clear, however, with a lack of studies that functionally assess HSPC fate at the single-cell level in vivo and analyze the direct effect of EPO on HSPCs, and not the effect of the surrounding niche. More specifically, the upregulation of erythroid associated genes in HSPC (LSK CD150$^+$ Flt3$^-$ CD48$^-$) in response to in vivo EPO has been observed with bulk transcriptomics (*Singh et al., 2018*), suggesting that HSPCs deviate their fate toward erythroid production. Recent single-cell RNA sequencing (scRNAseq) analysis observed different changes of lineage-associated gene expressions after in vivo EPO exposure of HSPCs (*Yang et al., 2017*; *Giladi et al., 2018*). As changes of gene expression do not necessarily result in cell fate modification (*Weinreb et al., 2020*), functional validations in vivo are necessary. In one study, such functional validation was performed using bulk transplantation of in vivo EPO-exposed HSPCs (LSK CD150$^+$ Flt3$^-$) (*Grover et al., 2014*). This study showed increased erythroid production and decreased myeloid cell production, concluding that EPO deviates the fate of HSPCs in favor of erythroid production. As EPO has also been shown to target hematopoietic niche cells (osteoblasts and osteocytes [*Li et al., 2015*; *Singbrant et al., 2011*], endothelial cells [*Ito et al., 2017*; *Singbrant et al., 2011*], adipocytes [*Alvarez and Noguchi, 2017*; *Zhang et al., 2014*], and mesenchymal stem cells [*Shiozawa et al., 2010*; *Tari et al., 2017*]), it remains however still unclear whether EPO acts directly on HSPCs, via their environment, or both.

It is now established that HSPCs encompass cells with different long-term reconstitution capacity after transplantation, as well as a heterogeneous output in terms of quantity (lineage bias) and type of cells (lineage restriction) (*Dykstra et al., 2007*; *Morita et al., 2010*; *Oguro et al., 2013*; *Sanjuan-Pla et al., 2013*; *Carrelha et al., 2018*; *Yamamoto et al., 2013*; *Lu et al., 2011*; *Verovskaya et al., 2013*). Recently, the HSPC compartment was subdivided into long-term hematopoietic stem cells (LT-HSC) and different multipotent progenitors (MPPs) (MPP1–4; *Wilson et al., 2008*; *Cabezas-Wallscheid et al., 2014*), with variation around the phenotypic definition (*Pietras et al., 2015*). Interestingly, HSPC composition responds to irradiation with HSC transiently self-renewing less and increasing their production of MPP2–3 (*Pietras et al., 2015*). In the case of EPO, conflicting results suggest that either HSC or MPP respond to EPO (*Giladi et al., 2018*; *Yang et al., 2017*; *Singh et al., 2018*), but the difference in HSPC definition and the lack of functional validation make it difficult to compare these studies. As HSPCs are functionally heterogeneous, and the current phenotypic definition partially captures this heterogeneity, a single-cell in vivo lineage-tracing approach is needed to assess whether EPO can influence HSPC fate decisions.

To analyze the functional effect of EPO on the differentiation of individual HSPCs (C-Kit$^+$ Sca1$^+$ CD150$^+$ Flt3$^-$) removing the effect of the niche, we here utilized cellular barcoding technology that allowed us to trace the progeny of hundreds of single HSPCs in vivo. By analyzing cellular barcodes in five mature hematopoietic lineages and HSPCs, we observed transient induction of high-output myeloid-erythroid-(MEK)-biased barcode clones compensated by myeloid-B-cell-dendritic cell (MBDC)-biased clones after ex vivo EPO exposure. scRNAseq of ex vivo EPO-exposed HSPCs revealed upregulation of erythroid associated genes in a subset of the compartment with overlap to gene signatures of MPP1 (C-Kit$^+$ Sca1$^+$ Flt3$^-$ CD150$^+$ CD48$^-$ CD34$^+$) and MPP2 (C-Kit$^+$ Sca1$^+$ Flt3$^-$ CD150$^+$ CD48$^+$) (*Cabezas-Wallscheid et al., 2014*; *Wilson et al., 2008*) and not of LT-HSCs (C-Kit$^+$ Sca1$^+$ Flt3$^-$ CD150$^+$ CD48$^-$ CD34$^-$). Transplantation of barcoded MPP2 confirmed their enrichment in ME-biased clones in response to EPO. Moreover, the increased contribution of biased HSPC clones to the mature cell lineages after EPO exposure did not match their frequency in HSPCs, indicating that they were differentiating more than self-renewing, a property associated to MPPs. The transient effect of EPO on HSPCs further corroborates an action of EPO on MPP1/2 rather than LT-HSCs. Altogether, our results are consistent with a model in which perturbations induce clonal remodeling of HSPC contributing to hematopoiesis, with biased MPPs transiently contributing more than LT-HSC. They also demonstrate a direct effect of EPO on MPPs after transplantation with implications for basic HSC research and therapeutic applications in the clinic.

## Results

### EPO exposure induces biases in single HSPCs

Given the debate surrounding which HSPC subset is responding to EPO, we decided to analyze the direct effect of EPO on the differentiation of HSPCs defined as C-Kit$^+$ Sca1$^+$ Flt3$^-$ CD150$^+$ (encompassing LT-HSC [C-Kit$^+$ Sca1$^+$ Flt3$^-$ CD150$^+$ CD48$^-$ CD34$^-$], MPP1 [C-Kit$^+$ Sca1$^+$ Flt3$^-$ CD150$^+$ CD48$^-$ CD34$^+$], and MPP2 [C-Kit$^+$ Sca1$^+$ Flt3$^-$ CD150$^+$ CD48$^+$]; *Cabezas-Wallscheid et al., 2014*; *Wilson et al., 2008*) at the single-cell level by cellular barcoding. To this purpose, we generated a new high-diversity lentiviral barcode library (LG2.2, 18,026 barcodes in reference list), consisting of random 20 nucleotides sequences positioned adjacent to the green fluorescent protein (GFP) gene, enabling the tracking of many individual cells in parallel. Using this LG2.2 library, we labeled single HSPCs (*Figure 1—figure supplement 1a*) with unique genetic barcodes as previously described (*Naik et al., 2013*), exposed them to EPO (1000 ng/ml) or PBS for 16 hr ex vivo, and transplanted around 2600 cells (mean 2684 cells ± 175 cells) of which around 10% barcoded cells into irradiated mice (*Figure 1a*). Note that HSPCs kept their sorting phenotype after ex vivo culture albeit a slight downregulation of C-Kit (*Matsuoka et al., 2011*) and upregulation of Flt3 (*Figure 1—figure supplement 1f*). At day 30 after transplantation, the earliest timepoint at which HSPCs produce simultaneously erythroid, myeloid, and lymphoid cells (*Boyer et al., 2019*), barcoded (GFP$^+$) erythroblasts (E; Ter119$^+$ CD44$^{+34}$), myeloid cells (M; Ter119$^-$ CD19$^-$ CD11c$^-$ CD11b$^+$), and B-cells (B; Ter119$^-$ CD19$^+$) (*Figure 1—figure supplement 1b,c,e*) were sorted from the spleen and their barcode identity assessed through PCR and deep sequencing. Note that bone and spleen had similar barcoding profiles (*Figure 1—figure supplement 2*). No difference in chimerism was observed between the EPO and control group in the spleen and blood, even when mTdTomato/mGFP donor mice were used to better assess the erythroid lineage (*Figure 1b and c*). On average, we detected around 80 barcodes per mouse, of which most were detected in several lineages (*Figure 1d*). Comparison of the numbers of barcodes producing each lineage showed that EPO exposure resulted in the same number of engrafting and differentiating cells as in control (*Figure 1d*). Notably, the number of erythroid restricted cells remained stable in the EPO group as compared to control (*Figure 1e*), indicating that the response to EPO is more complex than a direct instruction of erythroid-restricted HSPCs.

To quantify the effect of EPO on HSPC lineage biases, barcode-labeled HSPCs were classified based on the balance of their cellular output in the M, B, and E lineages. With this classification using a 10% threshold, cells classify, for example, as ME-biased if they have above 10% of their output in the M and E lineage and under 10% of their output in the B lineage (*Figure 1e*, other thresholds in *Figure 1—figure supplement 3b*). Interestingly, application of this classification revealed that although the proportion of lineage-biased HSPCs in the control and EPO groups was similar (*Figure 1f*), their contribution to the different lineages was increased by EPO exposure (*Figure 1g*). In the control group, balanced HSPCs (MBE) produced the majority of all lineages, as previously published (*Perié et al., 2015*). In the EPO group, ME- and MB-biased clones produced most cells of the analyzed lineages (*Figure 1g*). ME-biased HSPCs produced the majority of erythroid cells (57% ± 10%), MB-biased HSPCs produced the majority of B-cells (58% ± 36%), and ME- and MB-biased clones contributed the majority of myeloid cells (MB-biased 45% ± 38% and ME-biased 20% ± 13%, together 65% ± 25%). To test the significance of this effect, we used a permutation test that compares the effect size between the control and EPO groups to the one of all random groupings of mice (*Tak et al., 2019*). The contributions of the ME- and of the MB-biased HSPC classes to the different lineages were significantly different in the EPO and control groups (*Table 1*). These results were reproduced in an additional experiment (*Figure 1—figure supplement 3c–f*, *Supplementary file 1*). A lower EPO concentration (160 ng/ml) as well as an additional single injection of EPO (133 µg/kg) during transplantation gave similar results (*Figure 2*, *Figure 2—figure supplement 1a*, *Table 1*). Also at 6 weeks post-transplantation, similar results were obtained (*Figure 1—figure supplement 4*, *Supplementary file 1*). In summary, ex vivo EPO priming of HSPCs modified the output balance of HSPCs rather than the number of lineage-restricted and -biased cells. Balanced clones produced a smaller percentage of the mature cells; ME-biased HSPCs produced most of the erythroid cells and MB-biased HSPCs produced most of the B cells.

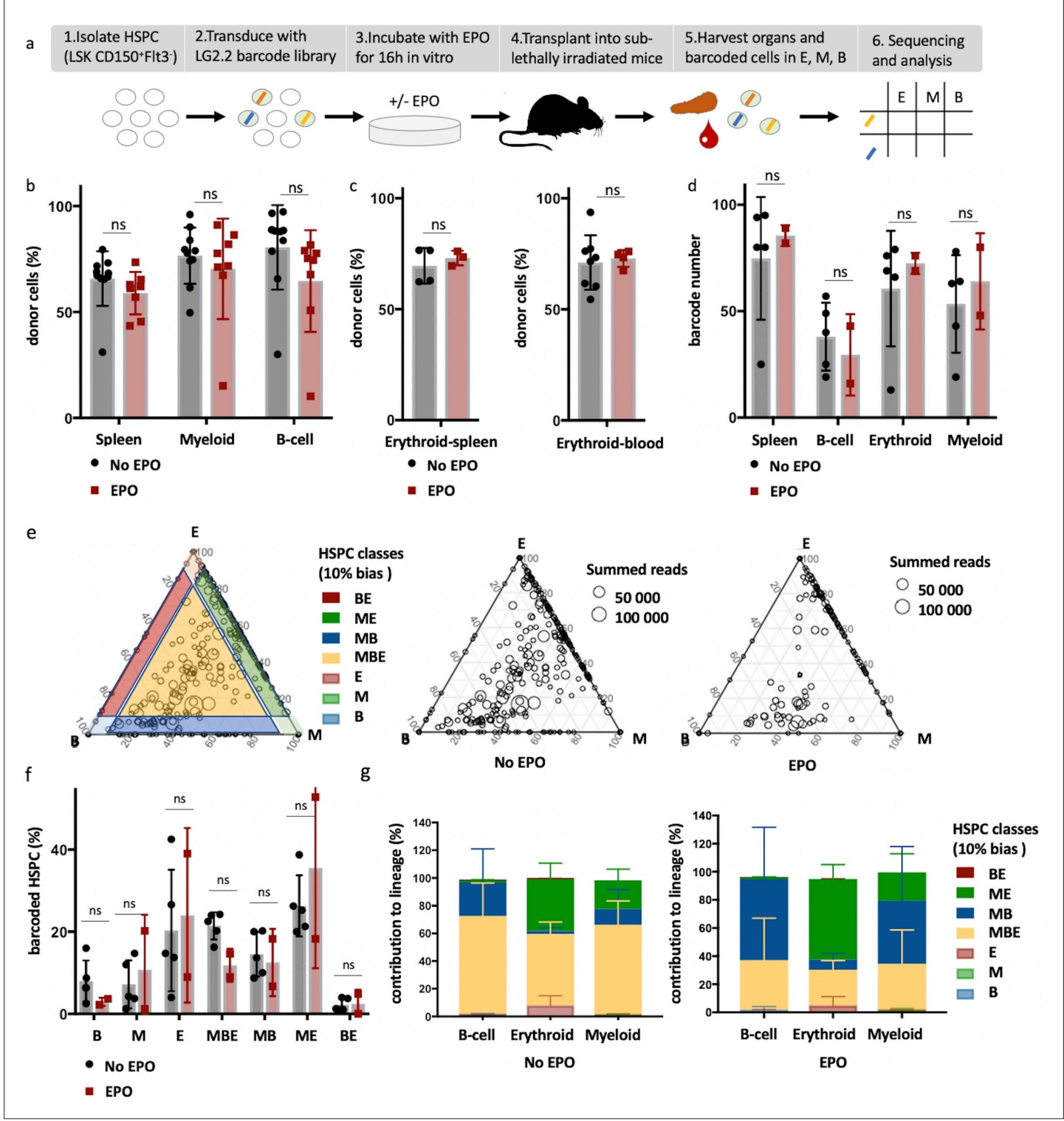

**Figure 1.** High-output ME- and MB-biased clones occur after erythropoietin (EPO) exposure and transplantation of hematopoietic stem and progenitor cells (HSPCs). (**a**) HSPCs were sorted from the bone marrow of donor mice, lentivirally barcoded, cultured ex vivo with or without 1000 ng/ml EPO for 16 hr, and transplanted into sublethally irradiated mice. At week 4 post-transplantation, the erythroid (E), myeloid (M), and B-cells (**B**) lineages were sorted from the spleen and processed for barcode analysis. (**b**) The percentage of donor-derived cells (CD45.1+) among the total spleen, myeloid cells (CD11b+) or B-cells (CD19+) in the spleen of control and EPO group. (**c**) To better assess chimerism in erythroid cells, mTdTomato/mGFP donor mice were used. The fraction of Tom+ cells among erythroid cells (Ter119+) in the spleen and blood in control and EPO group. (**d**) The number of barcodes retrieved in the indicated lineages at week 4 after transplantation in the control and EPO groups. (**e**) Triangle plots showing the relative abundance

*Figure 1 continued*

of barcodes (circles) in the E, M, and B lineage with respect to the summed output over the three lineages (size of the circles) for the control and EPO groups. (**f**) Tthe percentage of HSPCs classified by the indicated lineage bias using a 10% threshold for categorization. (**g**) Quantitative contribution of the classes as in (**f**) to each lineage. Shown are values from several animals (n = 8 EPO, n = 10 control in **b**, n = 3 EPO, n = 4 control in **c ,** spleen, n = 4 EPO, n = 8 control in **c ,** blood collected over five different experiments **d–g**, n = 5 for the control group and n = 2 for the EPO group collected over one experiment). For all bar graphs, mean and SD between mice are depicted. Statistical significance tested using Mann–Whitney *U*-test p=0,05 for (**b, c**). Statistical significance tested by permutation test for different subsets in (**g**) (see *Table 1*).

The online version of this article includes the following figure supplement(s) for figure 1:

**Figure supplement 1.** Gating strategies and hematopoietic stem and progenitor cell (HSPC) marker expression after lentiviral transduction and ex vivo culture with or without erythropoietin (EPO).

**Figure supplement 2.** Correlations in barcoding profiles of spleen and bone.

**Figure supplement 3.** Characterization of lineage biases after transplantation of erythropoietin (EPO)-exposed hematopoietic stem and progenitor cells (HSPCs).

**Figure supplement 4.** High-output ME- and MB-biased hematopoietic stem and progenitor cells (HSPCs) occur 6 weeks after transplantation of erythropoietin (EPO)-exposed HSPCs.

**Table 1.** Permutation testing of changes in clonality after transplantation of erythropoietin (EPO)-exposed hematopoietic stem and progenitor cells (HSPCs).

Same data as in *Figures 1–4 and 7*. HSPCs or multipotent progenitor 2 (MPP2) were cultured with different concentrations of EPO (160 ng/ml or 1000 ng/ml) for 16 hr, and when indicated a soluble dose of EPO (133 µg/kg) was injected together with barcoded HSPCs at the moment of transplantation. Barcodes in the erythroid (E), myeloid (M), B-lymphoid (B) lineage, dendritic cell (DC), and HSPCs were analyzed 4 weeks after transplantation and categorized by bias using a 10% threshold. For the data of *Figures 1, 2 and 7*, the output of MB and ME classified barcodes to the B, M, and E lineages was analyzed using a permutation test. For the data of *Figure 3*, the output of MBE and MB classified barcodes to the DC lineage was analyzed. For the data of *Figure 4*, the output of barcodes present in HSPCs to the B, M, and E lineages was analyzed using a permutation test. By permutating the mice of the control and EPO groups, the random distribution of this output was generated and compared to the real output difference between the control and EPO groups. A p-value was generated as in *Tak et al., 2019*.

| Figure | Condition | p-Value | | | |
| --- | --- | --- | --- | --- | --- |
| | | **MB in B** | **MB in M** | **ME in E** | **ME in M** |
| *Figure 1* | HSPCs 160 ng/ml | 0.02 | 0.04 | 0.02 | 0.04 |
| *Figures 1 and 2* | HSPCs 1000 ng/ml | 0.0075 | 0.0071 | 0.0071 | 0.012 |
| *Figure 2* | HSPCs 160 ng/ml + inj. | 0.01 | 0.016 | 0.012 | 0.011 |
| *Figure 2* | HSPCs 1000 ng/ml + inj. | 0.0018 | 0.0018 | 0.002 | 0.0025 |
| *Figure 7* | MPP2 1000 ng/ml | 0.006 | 0.008 | 0.004 | 0.004 |
| | | **MBE in DC** | **MB in DC** | | |
| *Figure 3* | HSPCs 1000 ng/ml | 0.07 | 0.0075 | | |
| | | **HSPC in B** | **HSPC in M** | **HSPC in E** | |
| *Figure 4* | HSPCs 160 ng/ml | 0.035 | 0.029 | 0.029 | |
| | HSPCs 1000 ng/ml | 0.008 | 0.06 | 0.01 | |
| | | **ME in E** | **ME in M** | | |
| *Figure 8* | HSPCs 160 ng/ml | 0.016 | 0.0025 | | |
| | HSPCs 1000 ng/ml | 0.00625 | 0.0038 | | |

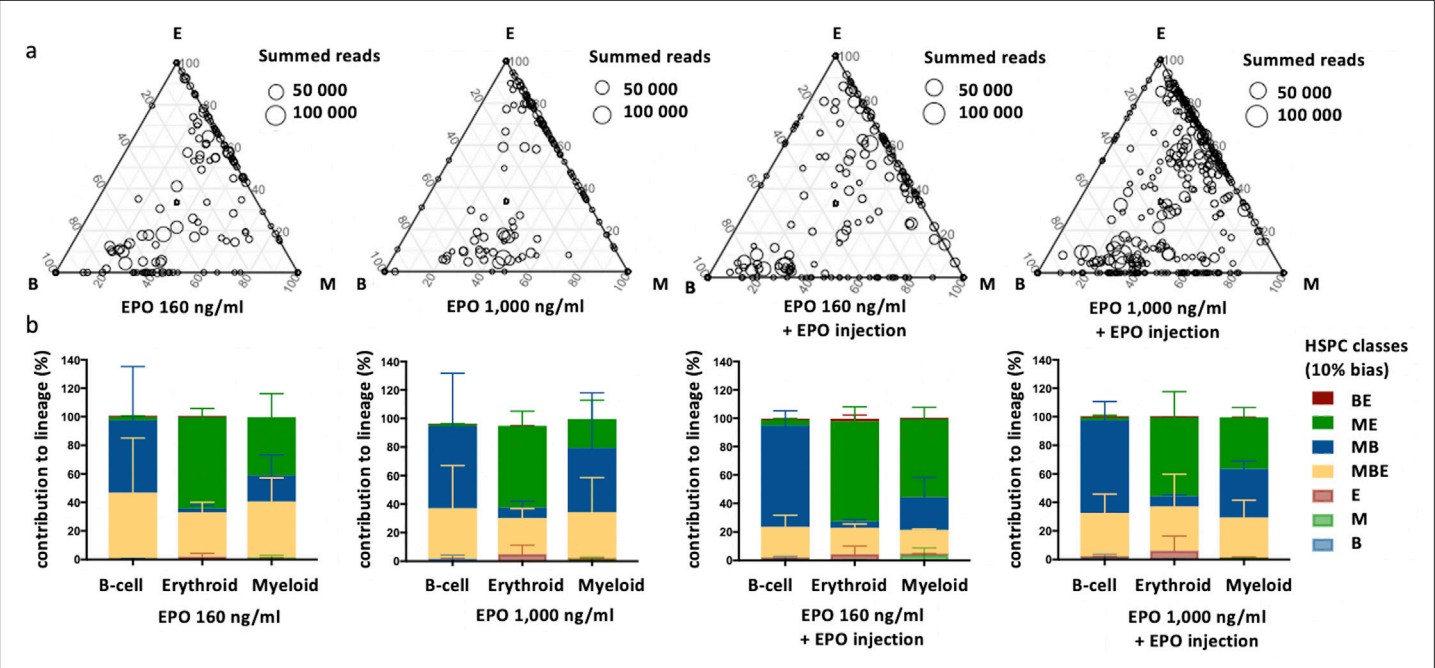

**Figure 2.** Effect of different erythropoietin (EPO) concentrations on hematopoietic stem and progenitor cell (HSPC) clonality after transplantation. Same protocol as in *Figure 1* but HSPCs were cultured with different concentrations of EPO (160 ng/ml or 1000 ng/ml) for 16 hr, and when indicated a single dose of EPO (133 μg/kg) was injected together with barcoded cells at the moment of transplantation. (**a**) Triangle plots showing the relative abundance of barcodes (circles) in the erythroid (E), myeloid (M), and B-lymphoid (B) lineage with respect to the summed output over the three lineages (size of the circles) for the different experimental groups as indicated. (**b**) The percentage of each lineage produced by the barcodes categorized by bias using a 10% threshold. Shown are values from several animals (n = 2 for 160 ng/ml, 1000 ng/ml, and 160 ng/ml + EPO injection, n = 4 for 1000 ng/ml + EPO injection [collected over four different experiments]). For all bar graphs mean and SD between mice are depicted. Statistical significance tested by permutation test for different subsets in (**b**) (see *Table 1*).

The online version of this article includes the following figure supplement(s) for figure 2:

**Figure supplement 1.** Variability in the effect of different erythropoietin (EPO) concentrations on clonality after hematopoietic stem and progenitor cell (HSPC) transplantation at different timepoints.

## Contribution of ME- and MB-biased HSPCs to the DC and MkP lineage

To further characterize the cells produced by the ME-biased and MB-biased HSPCs, we repeated our experimental setup including the analysis of the megakaryocyte and dendritic cell (DC) lineages (*Figure 3*). Megakaryocyte progenitors (MkP) were chosen as proxy for the production of platelets that are not suitable for barcode analysis. Barcoded (GFP$^+$) DCs (DC; Donor Ter119$^-$ CD19$^-$ CD11c$^+$ CD11b$^-$) and MkP (MkP; C-Kit$^+$ Sca-1$^-$ CD150$^+$ CD41$^+$) (*Figure 1—figure supplement 1c–e*) were sorted together with M, E, and B cells, 4 weeks after transplantation of control or EPO-exposed HSPCs (1000 ng/ml). In both groups, the majority of clones produced also DCs (*Figure 3a*). In the control group, balanced HSPCs produced the majority of DCs (65% ± 9%) (*Figure 3b*). However, in the EPO group, balanced HSPCs decreased their contribution to the DC lineage (36% ± 25%) and MB-biased HSPCs significantly increased their contribution (86% ± 43% EPO vs. 22% ± 11% control group) (*Figure 3b*, *Table 1*), thus, they were MBDC-biased HSPCs. In contrast, the ME-biased HSPCs produced few DCs in both groups (*Figure 3b*), indicating that ME-biased HSPCs are restricted both in their B and DC production compared to the M and E production.

The majority of the MkP production came from the ME-biased HSPCs in both groups (58% ± 21% control and 55% ± 14% EPO group, *Figure 3c and d*), indicating that ME-biased HSPCs were also MkP-biased HSPCs (thus MEK-biased). We did not detect a high contribution of MkP-restricted HSPCs (*Carrelha et al., 2018*; *Sanjuan-Pla et al., 2013*; *Rodriguez-Fraticelli et al., 2018*) to the MkP lineage (*Figure 3d*). Finally, as high EPO exposure has been linked to changes in macrophage numbers (*Theurl et al., 2016*; *Wang et al., 2018*; *Gilboa et al., 2017*; *Kuzmac et al., 2014*; *Ulyanova et al., 2016*; *Mausberg et al., 2011*; *Liao et al., 2018*; *Bretz et al., 2018*), we analyzed the contribution of control

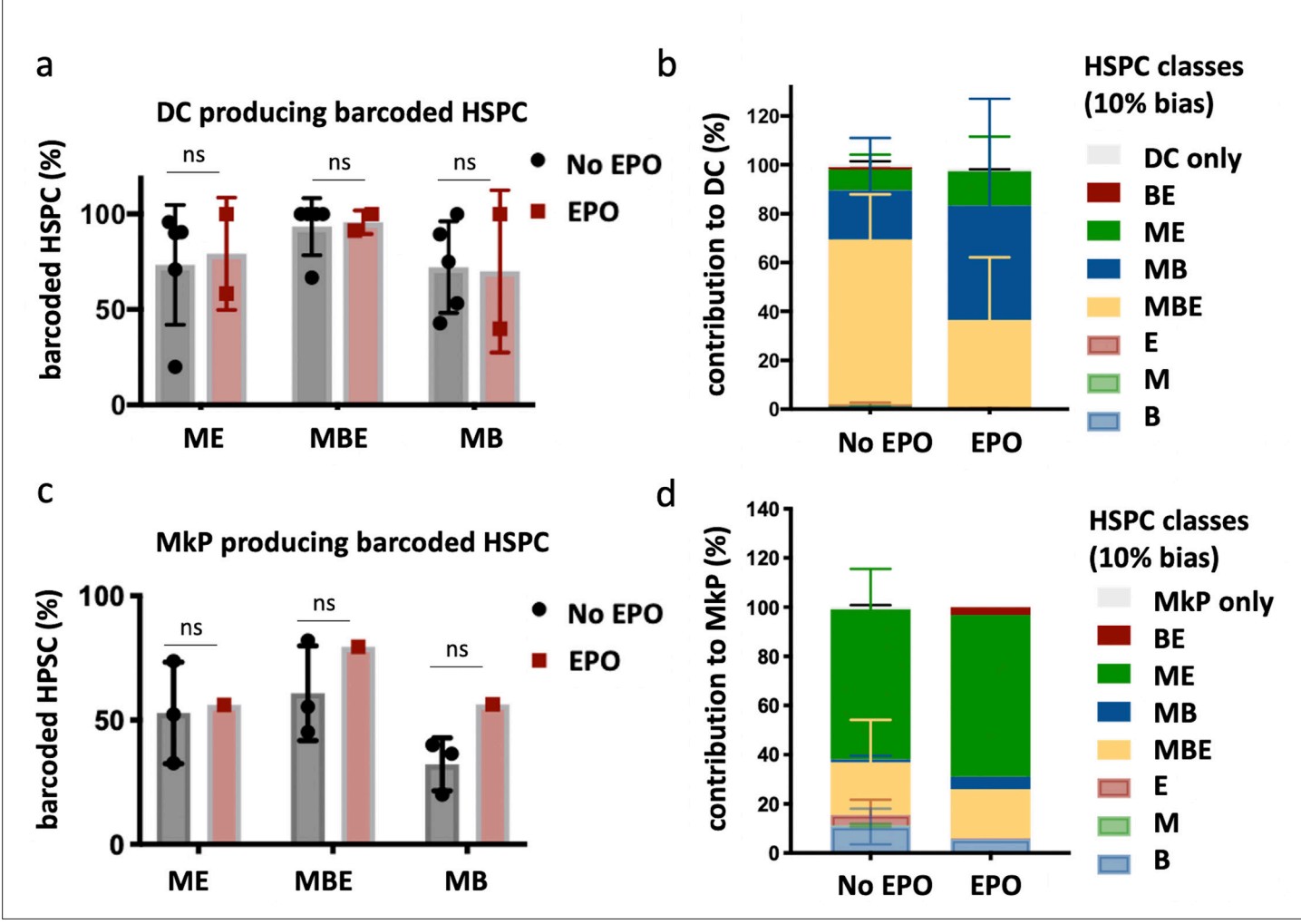

**Figure 3.** Production of dendritic cells (DCs) and megakaryocyte progenitors (MkP) by hematopoietic stem and progenitor cells (HSPCs) after erythropoietin (EPO) exposure and transplantation. In addition to the analysis of barcodes in the erythroid (E), the myeloid (M), and the B-cell (B) lineage, the DC lineage in spleen and MkP in bone marrow were added. (**a**) Percentage of barcoded HSPCs producing DC in the different HSPC categories (classification as in **Figure 2** based on the M, E, and B lineage only using a 10% threshold; the DC-only category was added). (**b**) The percentage of the DC lineage produced by the barcodes categorized by bias as in (**a**). (**c, d**) Representations as in (**a, b**) for barcode detection in MkP. Data is derived from a cohort with detailed myeloid sorting. The myeloid lineage was merged according to the percentage of total donor myeloid each subset contributed as in **Figure 2—figure supplement 1a** to allow classification as in (**a, b**) based on the M, E, and B lineage only using a 10% threshold. The MkP-only category was added. Shown are values from several animals (**a, b**, n = 5 for the control group and n = 2 for the EPO group; **c, d**, n = 3 for the control group and n = 1 for the EPO group [collected over two experiments]). For all bar graphs, mean and SD between mice are depicted. Statistical significance tested using Mann–Whitney *U*-test p=0.05 for (**a, c**). Statistical significance tested by permutation test for different subsets in (**b**) (see **Table 1**).

The online version of this article includes the following figure supplement(s) for figure 3:

**Figure supplement 1.** Production of macrophages (Ma), monocytes (Mo), neutrophils (Neu), eosinophils (Eo), and megakaryocyte progenitors (MkP) by hematopoietic stem and progenitor cells (HSPCs) after erythropoietin (EPO) exposure and transplantation.

and EPO-exposed HSPCs to the myeloid lineage in more detail, but could not detect changes in the percentage of the different myeloid subsets produced (**Figure 3—figure supplement 1a–c**).

## Effect of EPO on short-term HSPC self-renewal

In light of previous studies that suggested changes in HSPC proliferation after in vivo EPO exposure (**Cheshier et al., 2007**; **Dubart et al., 1994**; **Giladi et al., 2018**; **Yang et al., 2017**; **Singh et al., 2018**), we next explored if the short-term self-renewal capacity of HSPCs was impacted. To this end, we analyzed barcodes in bone marrow HSPCs in addition to the spleen E, M, and B lineages at week 4

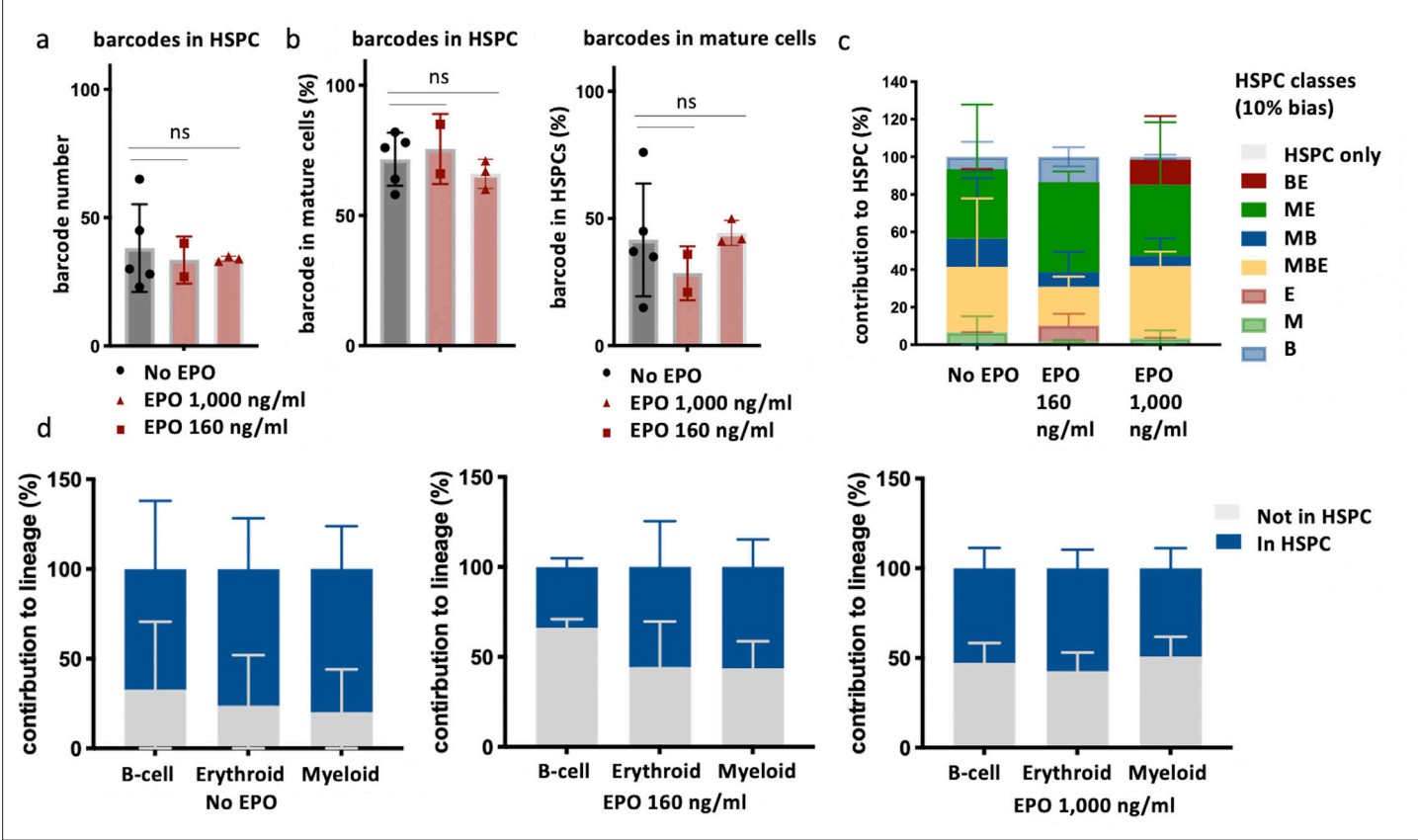

**Figure 4.** Overlap of barcodes in hematopoietic stem and progenitor cells (HSPCs) and mature cells after transplantation of erythropoietin (EPO)-exposed HSPCs. Same protocol as in *Figure 1* but HSPCs were cultured with two different concentrations of EPO (160 ng/ml or 1000 ng/ml) for 16 hr. In addition, HSPCs were sorted and subjected to barcode analysis. (**a**) The total number of barcodes found back in HSPCs. (**b**) The percentage of barcodes in the mature cell subsets also detected in HSPCs and the percentage of barcodes in HSPCs also detected in mature cells. (**c**) The percentage of the HSPC lineage contributed by barcodes categorized by bias as in *Figure 2* based on the myeloid (M), erythroid (E), and B-cells (B) lineage using a 10% threshold. (**d**) The percentage of each lineage produced by the barcodes color coded for presence (blue) and absence (gray) in HSPCs. Shown are values from several animals (n = 5 for the control group, n = 2 for the EPO 160 ng/ml group and n = 3 for the EPO 1000 ng/ml group). For all bar graphs, mean and SD between mice are depicted. Statistical significance tested using Mann–Whitney *U*-test p=0.05 for (**a**, **b**). Statistical significance tested by permutation test for different subsets in (**d**) (see *Table 1*).

after transplantation of control or EPO-exposed HSPCs (160 and 1000 ng/ml) (*Figure 4*). We reasoned that barcodes of HSPCs differentiating and short-term self-renewing (dividing to give rise to other HSPCs) after transplantation are detected in both compartments, while detection in only HSPCs or mature lineages indicates a prevalence of short-term self-renewal or differentiation respectively. Most of the barcodes detected in HSPCs overlapped with barcodes in the mature cells (*Figure 4b*, left) in both the control and two EPO groups, showing that most of the transplanted cells had given rise to other HSPCs and differentiated irrespective of the treatment. Some barcodes were only detected in mature cells (*Figure 4b*, right), indicating that some HSPCs had only differentiated or were below the limit of detection. These HSPCs were equally abundant in the control and two EPO groups (*Figure 4b*, right).

To analyze if different lineage biases correlated to different short-term self-renewal capacity, we analyzed the proportion of biased HSPC classes, as previously defined, within the HSPC compartment (*Figure 4c*). In the control group, balanced and ME-biased HSPCs contributed most to the HSPC reads (34% ± 36% MBE and 37% ± 34% ME-biased HSPCs), while barcodes of MB-biased HSPCs contributed less (15% ± 32%) (*Figure 4c*), a trend that has been previously described (*Kim et al., 2014*; *Aiuti et al., 2013*; *Carrelha et al., 2018*). Surprisingly, the pattern of contributions of different biased HSPC subsets to HSPC reads was unchanged in the EPO groups (*Figure 4c*), implying that the extent of short-term self-renewal was unchanged after ex vivo EPO exposure.

To study if the increased production of cells by the ME- and MB-biased HSPCs to the mature cells observed after ex vivo EPO exposure (*Figures 1–3*) correlated with short-term self-renewal capacity of HSPCs, we analyzed the contribution of barcodes detected or not in HSPCs to the E, M, and B lineages (*Figure 4d*). In the control group, the majority of mature cells were derived from barcodes also present in HSPCs. However, in both EPO groups, the contribution of barcodes detected in HSPCs to mature cells was significantly lower (*Figure 4d*, *Table 1*), implying that the increased contribution of biased HSPC classes to the mature cell lineages after ex vivo EPO exposure was most likely caused by cells differentiating more than short-term self-renewing.

## EPO exposure induces an erythroid program in a subgroup of HSPCs

To further characterize the effect of EPO exposure on HSPCs, we performed scRNAseq of barcoded C-Kit$^+$ Sca1$^+$ Flt3$^-$ CD150$^+$ cells after ex vivo culture in medium supplemented with EPO or PBS using the 10X Genomics Chromium platform. 1706 cells from control and 1595 cells from the EPO group passed our quality control. To compare the HSPCs injected with noncultured hematopoietic cells, we generated a reference map of 44,802 C-kit$^+$ cells from *Dahlin et al., 2018* and used published signatures as detailed in Materials and methods (*Pietras et al., 2015*; *Wolf et al., 2019*) to annotate this map (*Figure 5—figure supplement 1a, b and e, f*). Projection of our single-cell data on this map showed that both the control and the EPO-exposed HSPCs similarly overlapped with non-MPP4 LSK cells, according to their sorting phenotype (*Figure 5—figure supplement 1e and f*). These results indicate that neither the ex vivo culture itself nor the EPO treatment dramatically affected the global identity of the sorted HSPCs.

When comparing the EPO and control groups, we found 1176 differentially expressed genes (*Figure 5a*) and this number was significantly higher than the number expected due to chance (p-value=0.01) as assessed by permutation testing. Among the most upregulated genes in the EPO-exposed HSPCs were genes with erythroid association as *Hbb-bs*, *Erdr1*, *Wtap*, *Kmt2d*, or Nfia (*Starnes et al., 2009*), and GATA1 targets (*Abhd2*, *Cbx3*, *Kdelr2*, *Pfas*), cell cycle-related genes (*Tubb5*, *Hist1h2ap*), as well as genes previously described to be induced in HSPCs after in vivo EPO exposure, such as Bmp2k (*Shiozawa et al., 2010*) and Ifitm1 (*Giladi et al., 2018*); (*Figure 5a*). Genes involved in stem cell maintenance, such as *Serpina3g*, *Mecom, Txnip*, *Meis1*, *Pdzk1ip1*(*Giladi et al., 2018*), Sqstm1 (*Meenhuis et al., 2011*), Smad7 (*Blank et al., 2006*), and Aes (*Steffen et al., 2011*), were among the most downregulated genes in the EPO-exposed HSPCs (*Figure 5a*).

As our cellular barcoding data suggests that single HSPCs differ in their response to EPO, we assessed the heterogeneity of EPO responses at the transcriptomic level. UMAP-based visualization of the data suggested that a subgroup of EPO-exposed cells was transcriptomically distinct (*Figure 5b*), independently of the number of principal component analysis (PCA) components and genes used in the analysis (*Figure 5—figure supplement 1c*). To test this observation, we defined an EPO response signature based on differentially expressed genes between the EPO and control group. Plotting the expression of the EPO response signature at the single-cell level showed that the majority of the transcriptomic differences between the control and EPO group were indeed driven by this small subgroup of cells (*Figure 5c*). Reasoning that this subgroup contains the cells directly responding to EPO, we defined as EPO responders, cells in the 90th percentile of EPO response signature expression (*Figure 5c*) for subsequent analysis. Importantly, unsupervised clustering analysis of the data (*Figure 5—figure supplement 2a and b*) showed similar results. The genes encoding EPOR, as well as the alternative EPORs EphB4, CD131, and CRFL3, were equally expressed between the EPO responders, nonresponders, and control groups (*Figure 5—figure supplement 1d*). Reasoning that the EPO responders correspond to MEK-biased HSPCs, we also looked for potential MBDC-biased HSPCs but could not detect a subgroup of cells with upregulation of lymphoid-associated genes, suggesting that the MBDC bias is not a direct effect of EPO exposure but more an indirect effect. In summary, the scRNAseq analysis corroborated our functional barcoding data, showing that a subset of HSPCs can respond directly to EPO stimulation.

## EPO responder HSPCs overlap with MPP1 and MPP2 signatures

As our barcode analysis suggested that the effect of direct EPO exposure on HSPCs is caused by cells differentiating more than self-renewing, we next wanted to assess which of the HSPC subsets are the EPO responders in our scRNAseq dataset. We annotated the UMAP-based visualization of

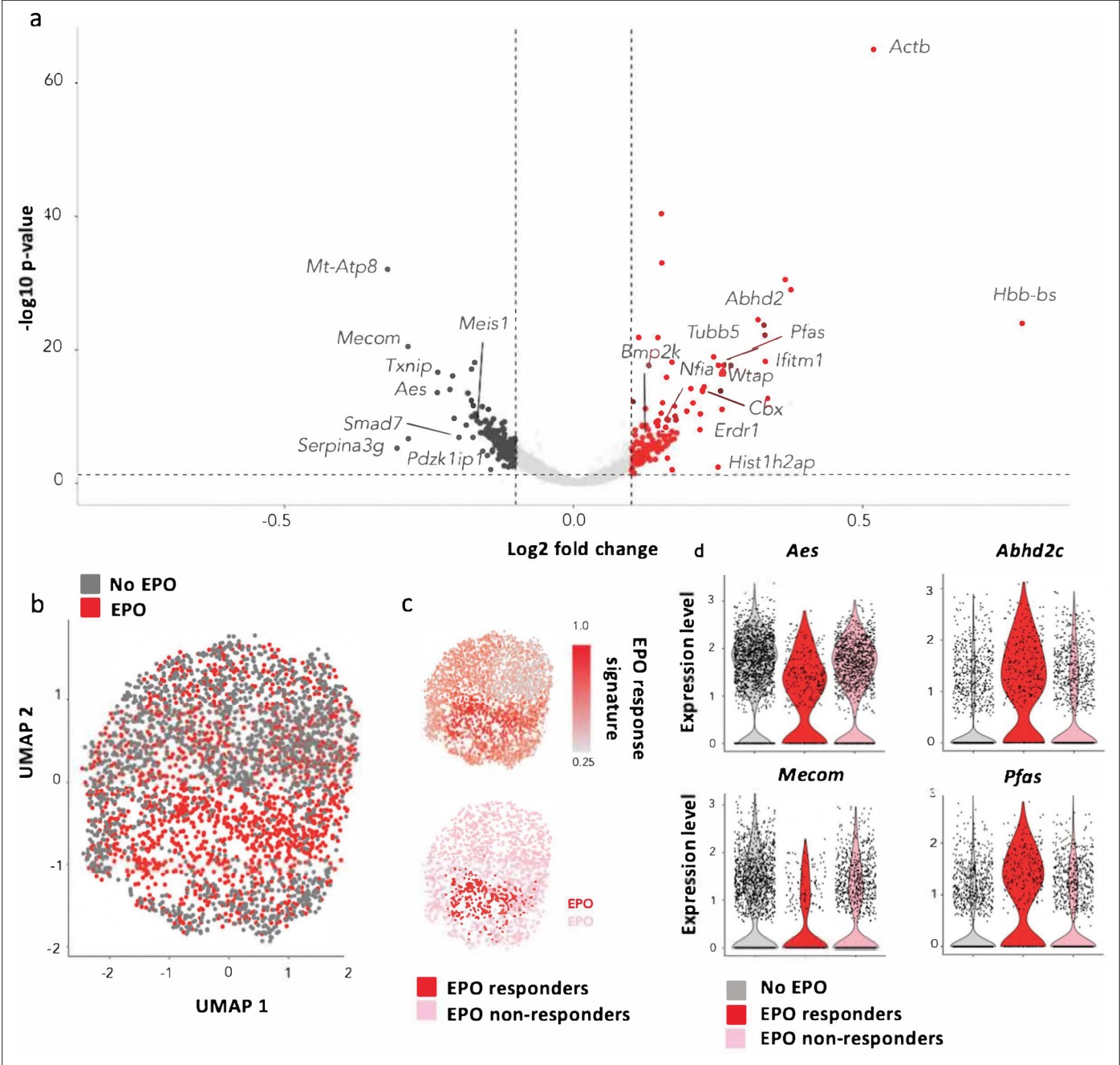

**Figure 5.** Characterization of erythropoietin (EPO)-exposed hematopoietic stem and progenitor cells (HSPCs) by single-cell RNA sequencing (scRNAseq). HSPCs were sorted, barcoded, and cultured ex vivo with or without 1000 ng/ml EPO for 16 hr and analyzed by scRNAseq using the 10X Genomics platform. 1706 cells from control and 1595 cells from the EPO group passed quality control. (**a**) Volcano plot of log2 fold change of the differentially expressed genes between control and EPO-exposed cells versus the adjusted p-value. Genes of interest are annotated. Differentially expressed genes were used to define an EPO response signature. (**b**) UMAP visualization of the EPO-exposed and control HSPCs. (**c**) The level of expression in the EPO-exposed HSPCs of the genes in the EPO response signature (top), and definition of the EPO responder and nonresponder subgroups using the 90th percentile expression of the EPO response signature from (**c**) (bottom). (**d**) The expression of the indicated genes in the control, EPO responder, and nonresponder subgroups as defined in (**c**). Genes that are significantly upregulated in the EPO responder group when compared to the control and nonresponder groups. Differential expression was assessed using a logistic regression testing approach, as implemented in Seurat. Figure supplements correspond to one 10× experiment of a pool of eight mice.

The online version of this article includes the following figure supplement(s) for figure 5:

**Figure supplement 1.** Single-cell RNA sequencing (scRNAseq) analysis of control and erythropoietin (EPO)-exposed hematopoietic stem and progenitor cells (HSPCs).

**Figure supplement 2.** Unsupervised clustering of control and erythropoietin (EPO)-exposed hematopoietic stem and progenitor cells (HSPCs).

our data with published signatures of the HSC (dormant HSCs [*Cabezas-Wallscheid et al., 2017*] and LT-HSC [*Wilson et al., 2015*]), MPP1 (*Cabezas-Wallscheid et al., 2017*), and MPP2 (*Pietras et al., 2015*) subsets included in our HSPC gate and analyzed its overlap with the previously defined EPO responder and nonresponder cells (*Figure 6a and b*). Relative to the control group and nonresponders of the EPO group, the EPO responders had a reduced expression of HSC gene signatures and increased expression of MPP1 and MPP2 signatures (*Figure 6a and b*). An annotation of the reference map generated from data of *Dahlin et al., 2018* likewise showed a low overlap of EPO responders with the most quiescent HSC subsets (*Figure 6c and d*). The independent analysis using unsupervised clustering further supported this result (*Figure 5—figure supplement 2c and d*). All in all, our scRNAseq analysis implied that, in line with our barcoding results, the HSPCs directly reacting to EPO are most likely MPP cells of the MPP1 and MPP2 subsets.

## EPO exposure induces ME biases in single MPP2

To confirm that MPP2 are a subset within HSPCs reacting directly to EPO as predicted by the scRNAseq analysis, we transplanted barcoded control or EPO-exposed (1000 ng/ml) MPP2 together with unbarcoded CD48$^-$ HSPCS (C-Kit$^+$ Sca1$^+$ Flt3$^-$ CD150$^+$ CD48$^-$) (*Figure 7—figure supplement 1a*) and analyzed their barcoded progeny in the E, M, and B lineages of the spleen at week 4 after transplantation (*Figure 7*). We found an equivalent engraftment as for the entire HSPC compartment and no difference between the EPO-treated and the control group (*Figure 7a and b*). Applying the same classification as in *Figure 1* to quantify the effect of EPO on MPP2 lineage biases, we observed that, as for the whole HSPC compartment, ME-biased cells contributed more to the M and E lineages (*Figure 7d and e*, other threshold in *Figure 7—figure supplement 1c*). Similarly to our data on whole HSPC compartment (*Figure 1g*), the proportion of the differently biased MPP2 was similar between the control and EPO groups (*Figure 7c*). This data confirms that the MPP2 population is enriched in HSPCs responding to EPO.

## Transient effect of EPO exposure

Finally, we reasoned that if EPO directly acts on MPPs 1/2 with short reconstitution capacity after transplantation rather than long-term repopulating HSC, then the EPO effect should be transient. To test this hypothesis, we repeated the experiment and analyzed barcodes in the E, M, and B lineages at 4 months after transplantation of control or EPO-exposed HSPCs (160 and 1000 ng/ml) (*Figure 8*). In the control group, as reported before (*Wu et al., 2018*), the chimerism at 4 months was higher and the number of barcodes detected was lower than at 1 month post-transplantation (*Figures 8c and d and 1b–d*). While ME-biased HSPCs still have significant higher contribution to the M and E lineages (*Figure 8a and b*, p-values in *Table 1*), the majority of cells in all lineages were produced by balanced HSPCs (*Figure 8a and b*), implying that the effect of direct EPO exposure on HSPCs is fading away. This confirms that the effect of direct EPO exposure on HSPCs is likely caused by MPP cells with a short reconstitution capacity after transplantation.

## Discussion

EPO is a key regulator of hematopoiesis and is classically considered to support the proliferation and survival of erythroid-committed progenitors. By analyzing the in vivo fate of hundreds of EPO-stimulated vs. untreated transplanted (c-Kit$^+$ Sca1$^+$ Flt3$^-$ CD150$^+$) HSPCs at the single-cell level, we established that EPO can change HSPC differentiation in the absence of an EPO-stimulated bone marrow microenvironment. Collectively our results yield two important conclusions: (i) EPO has a direct effect on HSPCs, that is, not solely due to the effects of EPO on the surrounding niche; and (ii) EPO directly remodels the clonal composition of HSPC by inducing fate-biased MPP and reducing the output of HSC.

Specifically, we observe that EPO induced MEK-biased (ME) and MBDC-biased (MB) HSPCs that produced the majority (>60%) of mature hematopoietic cells at 4 and 6 weeks after transplantation. In contrast, balanced HSPCs (MBE) had a reduced output of mature cells in response to EPO. The increased erythroid-associated gene signature in a subset of HSPCs after ex vivo EPO exposure suggests that EPO directly induces high-output MEK-biased HSPCs, which is indirectly compensated

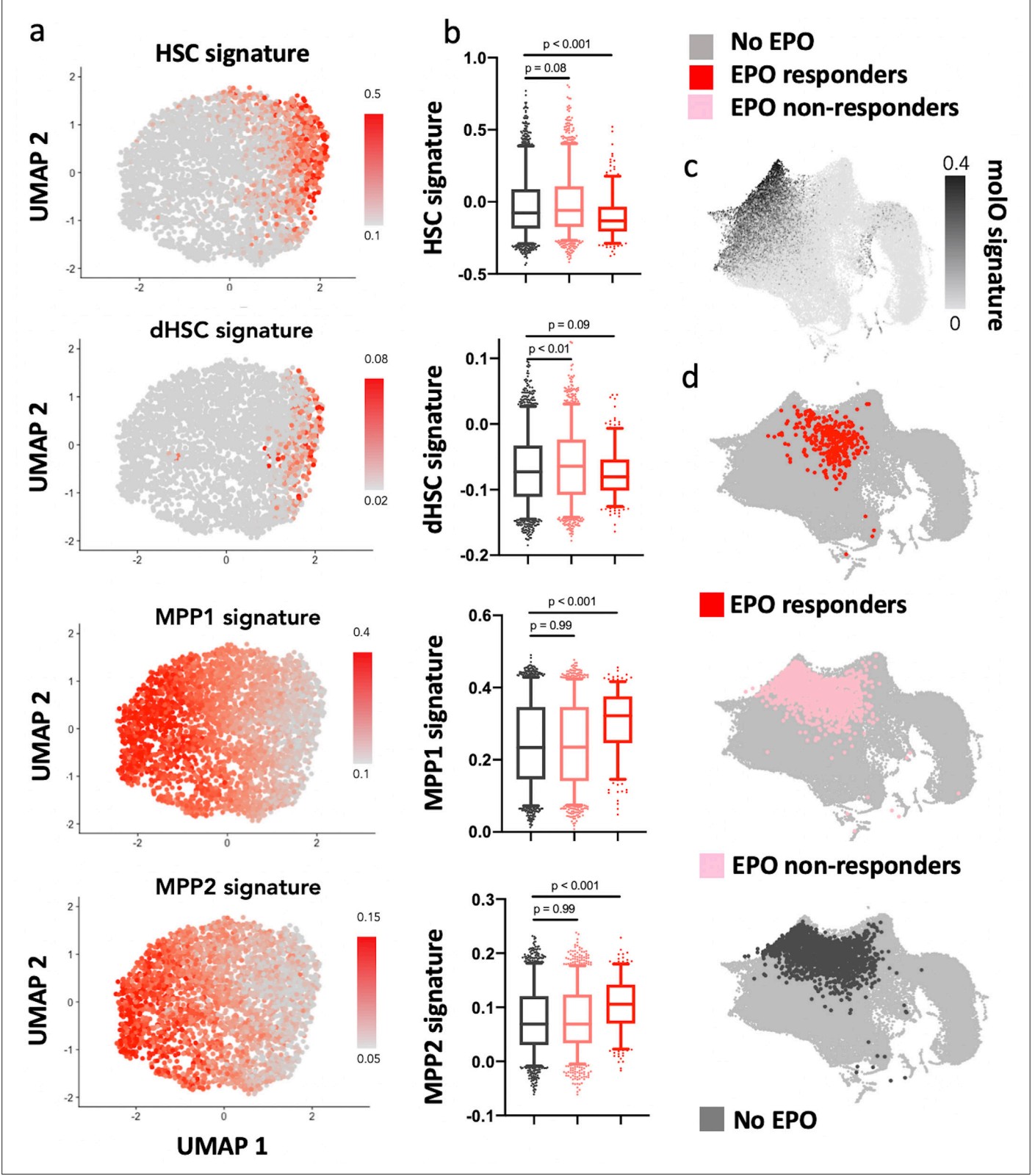

**Figure 6.** Erythropoietin (EPO) responders are multipotent progenitors, not hematopoietic stem cells (HSCs). Same protocol as in *Figure 5*. (**a**) Expression of published gene signatures of HSCs (dormant HSC [*Cabezas-Wallscheid et al., 2017*], molecular overlap [molO] HSC signature [*Wilson et al., 2015*]) and multipotent progenitors (MPPs) (MPP1 [*Cabezas-Wallscheid et al., 2017*]-2 [*Pietras et al., 2015*]) across the entire dataset (see Materials and methods). (**b**) Expression of the signatures from (**a**), across control, nonresponder, and EPO responder groups as defined in *Figure 5c*.

*Figure 6 continued on next page*

*Figure 6 continued*

Statistical comparisons made using a Kruskal–Wallis test with a Dunn's multiple comparisons post-hoc test. (**c**) Expression of the molO HSC signature on the published reference map (***Dahlin et al., 2018***). (**d**) Nearest-neighbor mapping of control, EPO responder, and nonresponder cells onto the published reference map (***Dahlin et al., 2018***).

for by the occurrence of high output of MBDC-biased HSPCs to maintain a balanced production of hematopoietic cells.

These biased clones had a higher propensity to differentiate than to self-renew, and their response to EPO was transient, suggesting that EPO-responsive cells are multipotent progenitors, and not LT-HSCs. This is supported by transcriptomic analysis showing that EPO responders express the MPP1/MPP2 gene signatures. Transplantation of barcoded EPO-exposed MPP2 confirmed their enrichment in ME-biased clones in response to EPO.

Similar to studies that assessed the effect of high systemic EPO exposure (***Tusi et al., 2018***; ***Yang et al., 2017***; ***Singh et al., 2018***; ***Giladi et al., 2018***), we found that MPPs, not HSCs, are responding

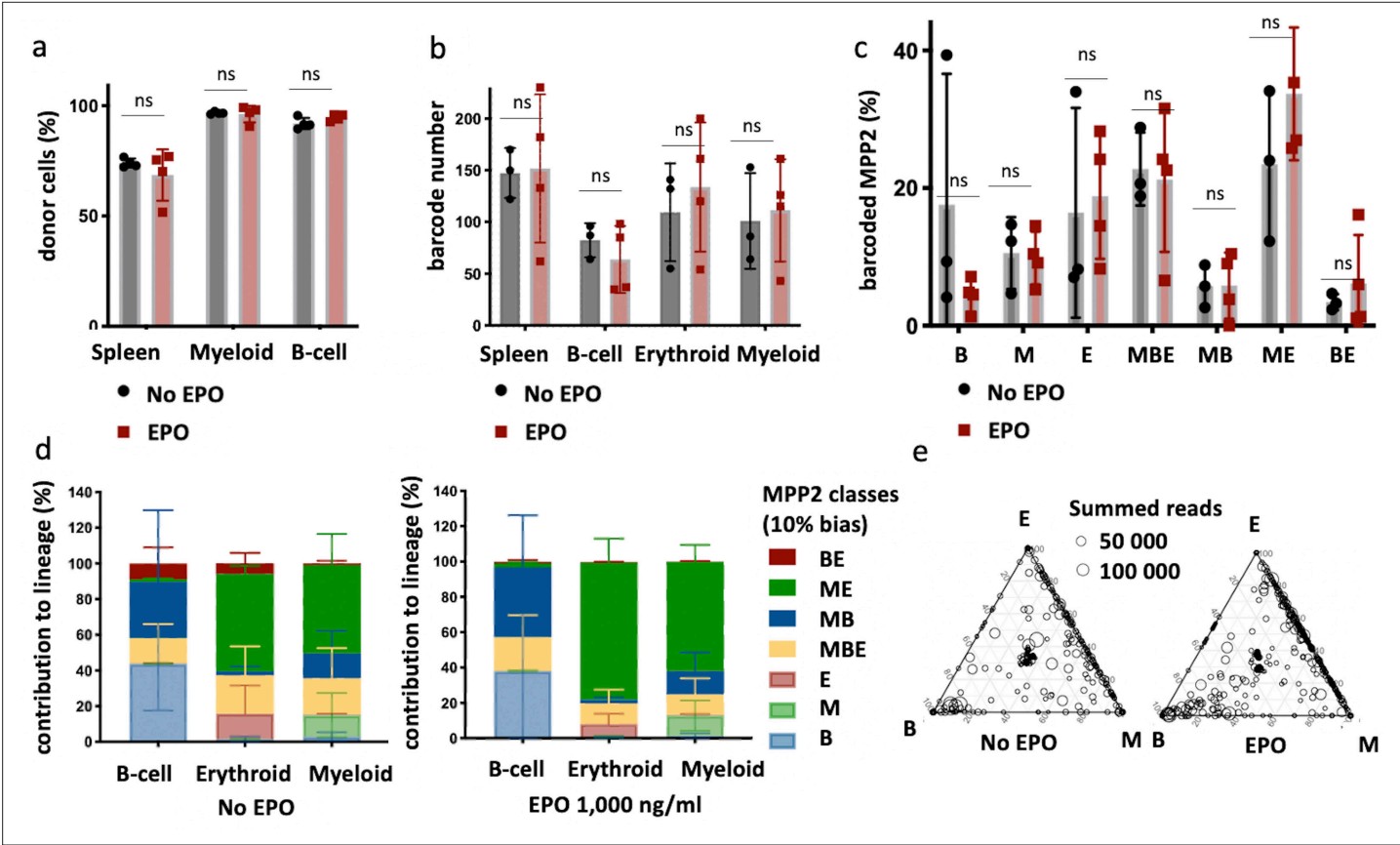

**Figure 7.** Multipotent progenitor 2 (MPP2) are enriched for ME-biased clones after erythropoietin (EPO) exposure and transplantation. MPP2 and CD48⁻ hematopoietic stem and progenitor cells (HSPCs) were sorted from the bone marrow of donor mice, MPP2 were lentivirally barcoded, and both populations cultured ex vivo with or without 1000 ng/ml EPO for 16 hr. After the culture, barcoded MPP2 and unbarcoded CD48⁻ HSPCs were mixed and transplanted into sublethally irradiated mice. At week 4 post-transplantation, the erythroid (E), myeloid (M), and B-cells (B) lineages were sorted from the spleen and processed for barcode analysis. (**a**) The fraction of donor cells among the indicated cell types in spleen. (**b**) Barcode number retrieved in the indicated lineage at 4 weeks after transplantation in the control and EPO 1000 ng/ml groups. (**c**) Percentage of MPP2s classified using a threshold of 10% in the experimental groups as indicated. (**d**) The percentage of each lineage produced by the MPP2 barcodes categorized by bias using a 10% threshold. (**e**) Triangle plots showing the relative abundance of barcodes (circles) in the erythroid (E), myeloid (M), and B-lymphoid (B) lineage with respect to the summed output over the three lineages (size of the circles). Shown are data from several mice (n = 3 for the control group and n = 4 for the EPO group). For all bar graphs, mean and SD between mice are depicted. Statistical significance tested using Mann–Whitney *U*-test p=0,05 for (**c–e**). Statistical significance tested by permutation test for different subsets in (**a**) (see ***Table 1***).

The online version of this article includes the following figure supplement(s) for figure 7:

**Figure supplement 1.** Characterization of lineage biases after transplantation of erythropoietin (EPO)-exposed multipotent progenitor 2 (MPP2).

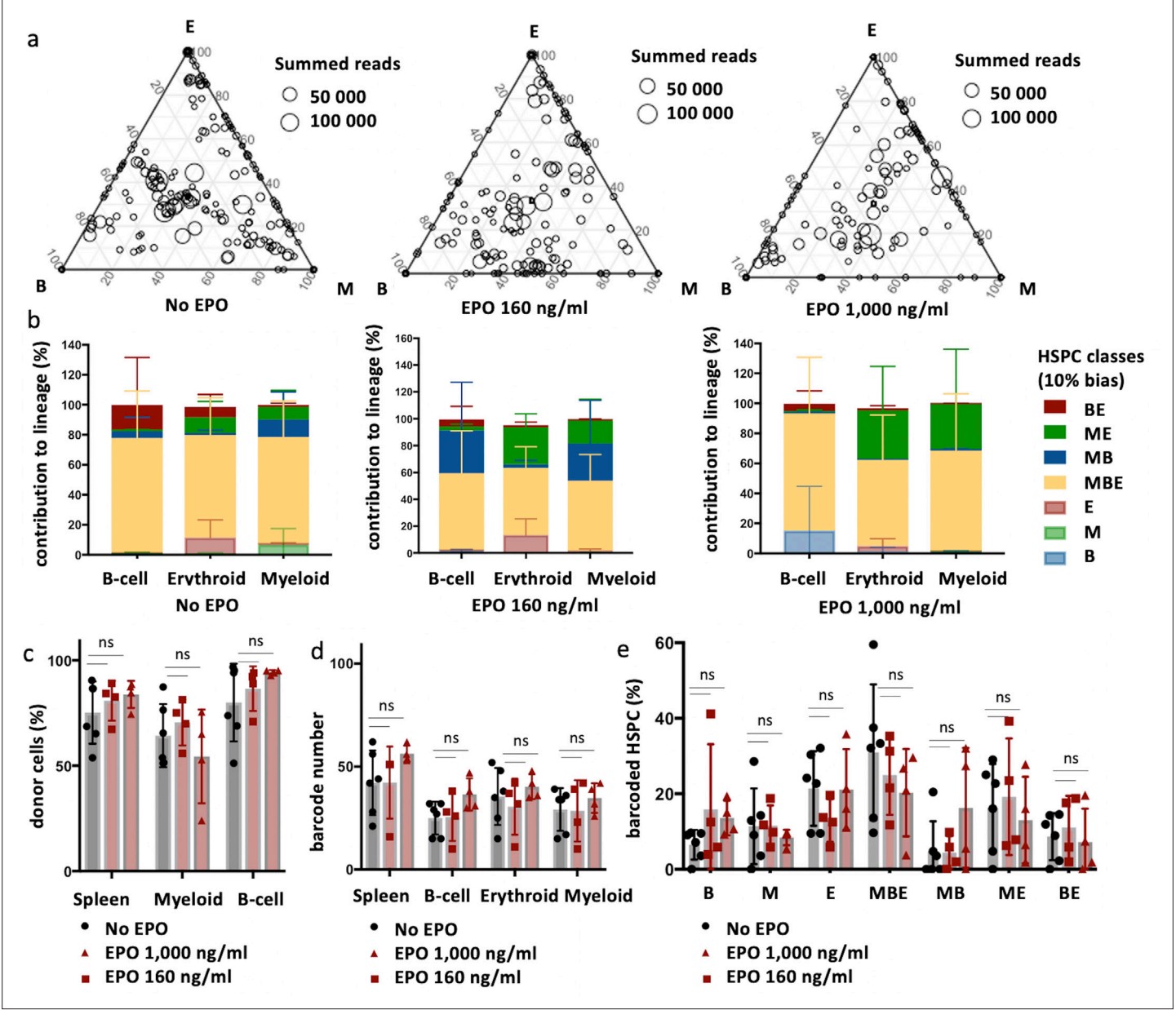

**Figure 8.** The effect of erythropoietin (EPO) on hematopoietic stem and progenitor cell (HSPC) clonality after transplantation is transient. Same protocol as in *Figure 1*, but barcodes in the erythroid (E), myeloid (M), and B-cell (B) lineage in spleen of individual mice sacrificed at month 4 post-transplantation were analyzed. (**a**) Triangle plots showing the relative abundance of barcodes (circles) in the E, M, and B lineage with respect to the summed output over the three lineages (size of the circles) for the different experimental groups as indicated. (**b**) The percentage of each lineage produced by the barcodes categorized by bias using a 10% threshold. (**c**) The fraction of donor cells among the indicated cell types in spleen. (**d**) Barcode number retrieved in the indicated lineage at month 4 after transplantation in the control, EPO 160 ng/ml, and the EPO 1000 ng/ml group. (**e**) Percentage of HSPCs classified using a threshold of 10% in the experimental groups as indicated. Shown are data from several mice (**c**, n = 5 for the control group and n = 4 for each EPO group, **a, b, d, e**, n = 6 for the control group and n = 4 for each EPO group [collected over two experiments]). For all bar graphs, mean and SD between mice are depicted. Statistical significance tested using Mann–Whitney *U*-test p=0.05 for (**c–e**). Statistical significance tested by permutation test for different subsets in (**b**) (see *Table 1*).

to EPO. The occurrence of myeloid, megakaryocytic, and erythroid gene expression in MPP1 after bleeding (*Yang et al., 2017*) is in line with our findings. Previously, long-term EPO exposure in Tg6 transgenic mice did however not change the in vitro differentiation outcome of MPP2 (*Singh et al., 2018*). Furthermore, we did not detect a fate deviation toward erythroid production at the expense of myeloid production as seen for in vivo EPO-exposed HSPCs after transplantation (*Grover et al.,*

*2014*). These differences could be due both to the duration and route of EPO exposure, as well as the indirect effects of systemic EPO exposure through other cells, for example, from the bone marrow niche.

In addition, our data shows that direct cytokine stimulation leads to a clonal remodeling of the HSPC compartment, with a transient increase in the contribution of fate-biased MPPs. Without longitudinal barcoding data within the same animal, we cannot distinguish if EPO is transiently changing the fate and outcome of the same HSPCs over time or if EPO is pushing the differentiation of some HSPCs that will be replaced by more balanced and stable HSPCs.

Different studies have suggested that the behavior of transplanted HSPC differs from native HSPC (*Sun et al., 2014*; *Busch et al., 2015*). Transplantation seems to favor the long-term output from HSC, whereas steady-state hematopoiesis is maintained more by MPPs' contribution than LT-HSC (*Sun et al., 2014*; *Busch et al., 2015*; *Schoedel et al., 2016*). Our work is in line with a model in which MPPs are a highly malleable cell population that can rapidly respond to changing demands for new cells, such as transplantation or infection (*Pietras et al., 2015*).

The direct effect of EPO on MPPs we described here could be one of the factors underlying the development of adverse side effects and comorbidities during long-term EPO use in the clinics and associations of high EPO levels with leukemias (*Rainville et al., 2016*; *Ma et al., 2010*; *Halawi et al., 2017*). To translate these results to the clinics and understand the side effect of EPO treatment, further work is required to determine if HSPCs and erythroid progenitors like CFU-E are responding to the same dose and duration of EPO exposure.

# Materials and methods

## Key resources table

| Reagent type (species) or resource | Designation | Source or reference | Identifiers | Additional information |
|---|---|---|---|---|
| Strain, strain background (*Mus musculus*) | C57BL/6J CD45.1⁺ | Jackson Laboratory | B6.SJL-*Ptprc*ᵃ *Pepc*ᵇ/BoyJ, Stock# 002014, B6 Cd45.1 | Male |
| Strain, strain background( *M. musculus*) | C57BL/6J CD45.2⁺ | Jackson Laboratory | C57BL/6J, Stock# 000664, B6 | Male |
| Strain, strain background (*M. musculus*) | *Rosa26CreER^{T2}*;*mT/mG* | Jackson Laboratory | STOCK *Gt(ROSA)26Sor^{tm4(ACTB-tdTomato,-EGFP)Luo}*/J, Stock# 007576, mT/mG, mTmG | Male |
| Strain, strain background (*Escherichia coli*) | ElectroMAX Stbl4 Competent Cells | Thermo Fisher Scientific | Cat# 11635018 | |
| Recombinant DNA reagent | pRRL-CMV-GFP plasmid (*Dull et al., 1998*) | PMID:9765382 | | Ton Schumacher lab, NKI, Amsterdam |
| Cell line (human) | HEK293T cells | Other | | Philippe Benaroch lab, Institute Curie, Paris |
| Recombinant DNA reagent | p8.9-QV | Other | | Philippe Benaroch lab, Institute Curie, Paris |
| Recombinant DNA reagent | pVSVG | Other | | Philippe Benaroch lab, Institute Curie, Paris |
| Chemical compound, drug | Anti-CD117 magnetic beads | Miltenyi | Cat# 130-091-224; RRID:AB_2753213 | |
| Chemical compound, drug | Propidium iodide | Sigma | Cat# 81845 | |
| Chemical compound, drug | StemSpanMedium SFEM | STEMCELL Technologies | Cat# 9650 | |
| Chemical compound, drug | Mouse recombinant SCF | STEMCELL Technologies | Cat# 78064.2 | |
| Chemical compound, drug | Eprex, erythropoietin alpha | Janssen | | |

*Continued on next page*

*Continued*

| Reagent type (species) or resource | Designation | Source or reference | Identifiers | Additional information |
|---|---|---|---|---|
| Chemical compound, drug | Anti-biotinylated beads | Miltenyi | Cat# 130090485; RRID:AB_244365 | |
| Antibody | Anti-Ter119-biotin (rat, monoclonal) | BD Biosciences | Cat# 553672, clone TER119; RRID:AB_394985 | (1:100) |
| Antibody | Anti-cd45.1-PE (mouse, monoclonal) | BD Biosciences | Cat# 553776, clone A20; RRID:AB_395044 | (1:100) |
| Antibody | Anti-Ter119-PE-Cy7 (rat, monoclonal) | BD Biosciences | Cat# 557853, clone TER119; RRID:AB_396898 | (1:100) |
| Antibody | Anti-cd11c-APC (hamster, monoclonal) | eBioscience | Cat# 17-0114-82, clone N418; RRID:AB_469346 | (1:100) |
| Antibody | Anti-cd19-APC-Cy7 (rat, monoclonal) | BD Biosciences | Cat# 557655, clone ID3; RRID:AB_396770 | (1:100) |
| Antibody | Anti-cd11b-PerCP-Cy5.5 (rat, monoclonal) | eBioscience | Cat# 45-0112-82, clone M1/70; RRID:AB_953558 | (1:100) |
| Antibody | Anti-cd117-APC (rat, monoclonal) | BioLegend | Cat# 105812, clone 2B8; RRID:AB_313221 | (1:100) |
| Antibody | Anti-cd135-PE (rat, monoclonal) | eBioscience | Cat# 12 135182, clone A2F10; RRID:AB_465859 | (100) |
| Antibody | Anti-cd135-PE-Cy5 (rat, monoclonal) | Life Technologies | Cat# 15_1351_82, clone A2F10; RRID:AB_494219 | (1:100) |
| Antibody | Anti-Sca1-PacificBlue (rat, monoclonal) | BioLegend | Cat# 122520, clone D7; RRID:AB_2143237 | (1:200) |
| Antibody | Anti-cd150-PE-Cy7 (rat, monoclonal) | BioLegend | Cat# 115914, clone TC15-12F12.2; RRID:AB_439797 | (1:100) |
| Antibody | Anti-cd44-PE (rat, monoclonal) | BD Biosciences | Cat# 553134, clone IM7; RRID:AB_394649 | (100) |
| Antibody | Anti-cd41-BV510 (rat, monoclonal) | BD Biosciences | Cat# 740136, clone MVVREG30; RRID:AB_2739892 | (1:100) |
| Antibody | Anti-Siglec-F-PE-CF594 (rat, monoclonal) | BD Biosciences | Cat# 562757, clone E50-2440; RRID:AB_2687994 | (1:200) |
| Antibody | Anti-Ly6g-BV510 (rat, monoclonal) | BioLegend | Cat# 127633. clone 1A8; RRID:AB_2562937 | (1:200) |
| Antibody | Anti-cd115-PE (rat, monoclonal) | BioLegend | Cat# 135505, clone AFS98; RRID:AB_1937254 | (1:200) |
| Antibody | Anti-cd48- APC-Cy7 (hamster, monoclonal) | BD Biosciences | Cat# 561242 clone HM48-1; RRID:AB_10644381 | (1:100) |
| Chemical compound, drug | Viagen Direct PCR Lysis Reagent (cell) | Euromedex | Cat# 301C | |
| Chemical compound, drug | Proteinase K Solution RNA grade | Invitrogen | Cat# 25530-049 | |
| Sequence-based reagent | top-LIB | This paper | PCR primer | TGCTGCCGTCAACTAGA ACA |
| Sequence-based reagent | bot-LIB | This paper | PCR primer | GATCTCGAATCAGGCGCTTA |
| Sequence-based reagent | PCR2-Read1-plate-index-forward | This paper | PCR primer | ACACTCTTTCCCTACACGACGCTCTTCCGAT CTNNNNCTAGAACACTCGAGATCAG |
| Sequence-based reagent | PCR2-Read2-reverse | This paper | PCR primer | GTGACTGGAGTTCAGACGTGTGCTCTTCCGAT CGATCTCG AATCAGGCGCTTA |
| Sequence-based reagent | PCR3-P5-forward | This paper | PCR primer | AATGATA CGGCGACCACCGAGATCTACACTCTTTCCC TACACGACGCTCTTCCGATCT |

*Continued on next page*

*Continued*

| Reagent type (species) or resource | Designation | Source or reference | Identifiers | Additional information |
|---|---|---|---|---|
| Sequence-based reagent | PCR3-P7-sample-index-reverse | This paper | PCR primer | CAAGCAGAAGACGGCATACGAGANNNNNNN GTGACTGGAGTTCAGA CGTGCTCTTCCGATC |
| Commercial assay or kit | Agencourt AMPure XP system | Beckman Coulter | Cat# A63881 | |
| Commercial assay or kit | Chromium Single Cell 3' Reagent Kits v2 Chemistry | 10X Genomics | | |
| Software, algorithm | R-3.4.0 | Other | | R Development Core Team (2019) http://www.R-project.org |
| Software, algorithm | GraphPad Prism version 8.0 for Mac | GraphPad | RRID:SCR_002798 | GraphPad Software, La Jolla, CA, https://www.graphpad.com |
| Software, algorithm | XCALIBR | Other | | https://github.com/NKI-GCF/xcalibr; *Netherlands Cancer Institute - Genomics Core Facilty, 2015* |
| Software, algorithm | Cellranger v3 | 10X Genomics | RRID:SCR_017344 | |
| Software, algorithm | Seurat v3 | doi:10.1016/j. cell.2019.05.031 | RRID:SCR_007322 | |

## Mice

Male C57BL/6J CD45.1⁺, C57BL/6J CD45.2⁺, and *Rosa26CreER^{T2};mT/mG* mice from Jackson Laboratory or bred at Institute Curie aged between 7 and 13 weeks were used in all experiments. All procedures were approved by the responsible national ethics committee (APAFIS# 10955–201708171446318v1).

## Barcode library, barcode reference list, and lentivirus production

The LG2.2 barcode library is composed of a DNA stretch of 180 bp with a 20 bp 'N'-stretch. DsDNA was generated by 10 PCR rounds and cloned into the XhoI-EcoRI site of the lentiviral pRRL-CMV-GFP plasmid (*Dull et al., 1998*). Subsequently, ElectroMaxStbl4 cells were transformed, and >10,000 colonies picked for amplification by Maxiprep. To create the barcode reference list (https://github.com/ PerieTeam/Eisele-et-al.-; *Eisele et al., 2022*), barcode plasmids were PCR amplified twice in duplicate and sequenced as described below. Sequencing results were filtered for barcode reference list generation as previously described in *Naik et al., 2013*. Lentiviruses were produced by transfecting the barcode plasmids and p8.9-QV and pVSVG into HEK293T cells in DMEM-GlutaMAX (Gibco) supplemented with 10% FCS (Eurobio), 1% MEM NEAA (Sigma), and 1% sodium pyruvate (Gibco) using polyethyleneimine (Polysciences). Supernatant was 0.45 μm filtered, concentrated by 1.5 hr ultracentrifugation at 31,000 × *g*, and frozen at –80°C. HEK293T were tested for mycoplasma contamination every 6 months.

## HSPC and MPP2 isolation, barcoding, EPO treatment, and transplantation

Isolation and labeling of cells with the barcoding library were performed as described in *Naik et al., 2013*. Briefly, after isoflurane anesthesia and cervical dislocation, bone marrow cells were isolated from femur, tibia, and iliac bones of mice by flushing, and C-Kit⁺ cells were enriched with anti-CD117 magnetic beads on the MACS column system (Miltenyi). Cells were stained for C-Kit, Flt3, CD150, Sca-1 (Key resources table) propidium iodide (PI) (Sigma) (1:5000), and if appropriate CD48. Lineage staining was not performed after C-Kit+ MACS enrichment for transplantation cohorts. For HSPC cohorts, HSPCs (*Figure 1—figure supplement 1a*) were sorted and transduced with the barcode library in StemSpanMedium SFEM (STEMCELL Technologies) with 50 ng/ml mSCF (STEMCELL Technologies) through 1.5 hr of centrifugation at 300 × *g* and 4.5 hr incubation at 37°C to obtain 10% barcoded cells. After transduction, cells were incubated with human recombinant EPO (Eprex, erythropoietin alpha, Janssen) at a final concentration of 1000 or 160 ng/ml or PBS for 16 hr at 37°C. After the incubation, the cells were transplanted by tail vein injection in recipient mice 6 Gy sublethally irradiated 3 hr before on a CIXD irradiator. Mice were allocated to groups of 4–5 mice for each condition randomly without masking. When indicated, cells were injected together with additional EPO (133 μg/

kg). On average, 2600 cells (mean 2684 cells ± 175 cells) were injected in the tail vein of each mouse. For the MPP2 cohort, MPP2 and CD48⁻ HSPCs (*Figure 7—figure supplement 1a*) were sorted. Both populations were cultured alike, but only MPP2 were transduced with the barcode library and treated with 1000 ng/ml recombinant EPO as described above. After the culture, barcoded MPP2 and unbarcoded CD48⁻ HSPCs were mixed at a ratio of 32/45 (to be as close as possible to the original ratio of both populations in the HSPCs) and transplanted as described above. A FACSAria (BD Biosciences) was used for sorting. FACSDiva software (BD Biosciences) was used for measurements and FlowJo v.10 (TreeStar) for analysis.

## Cell progeny isolation for barcode analysis

Spleens were mashed and both blood and spleen cells were separated based on Ter119 using a biotinylated anti-Ter119 antibody (Key resources table) and anti-biotinylated beads on the MACS column system (Miltenyi). Ter119⁺ cells were stained for Ter119 and CD44 (*Chen et al., 2009*). Ter119⁻ cells were stained for CD45.1 CD11b, CD11c, CD19, and, if appropriate, CD115, Siglec-F, and Ly6G (Key resources table). Bone marrow cells were flushed from bones and enriched for C-Kit⁺ cells as above. When appropriate, the C-Kit⁻ fraction was further separated based on Ter119 and stained as above. C-Kit⁺ cells were stained for C-Kit, Flt3, CD150, Sca-1, and, if appropriate, CD41 (Key resources table), and PI (1:5000) as described above. For analyzed and/or sorted populations, see *Figure 1—figure supplement 1*. Populations were only sorted for mice with an engraftment (donor cells percentage) of above 5% in spleen, bone, and blood.

## Lysis, barcode amplification, and sequencing

Sorted cells were lysed in 40 µl Viagen Direct PCR Lysis Reagent (cell) (Euromedex) supplemented with 0.5 mg/ml Proteinase K Solution RNA grade (Invitrogen) at 55°C for 120 min, 85°C for 30 min, and 95°C for 5 min. Samples were then split into two replicates, and a three-step nested PCR was performed to amplify barcodes and prepare for sequencing. The first step amplifies barcodes (top-LIB [5′TGCTGCCGTCAACTAGA ACA-3′] and bot-LIB [5′GATCTCGAATCAGGCGCTTA-3′]). A second step adds unique 4 bp plate indices as well as Read 1 and 2 Illumina sequences (PCR2-Read1-plate-index-forward        5′ACACTCTTTCCCTACACGACGCTCTTCCGATCTNNNNCTAGAACACTCGAGATCAG3′ and    PCR2-Read2-reverse    5′GTGACTGGAGTTCAGACGTGTGCTCTTCCGAT    CGATCTCGAATC AGGCGCTTA3′). In a third step, P5 and P7 flow cell attachment sequences and one of 96 sample indices of 7 bp are added (PCR3-P5-forward 5′AATGATA CGGCGACCACCGAGATCTACACTCTTTC CCTACACGACGCTCTTCCGATCT3′ and PCR7-P7-sample-index-reverse 5′CAAGCAGAAGACGGCA TACGAGANNNNNNNGTGACTGGAGTTCAGA CGTGCTCTTCCGATC3′) (PCR program: hot start 5 min 95°C, 15 s at 95°C; 30 s at 57.2°C; 30 s at 72°C, 5 min 72°C, 30 [PCR1-2] or 15 cycles [PCR 3]). Both index sequences (sample and plate) were designed based on *Faircloth and Glenn, 2012* such that sequences differed by at least 2 bp (https://github.com/PerieTeam/Eisele-et-al.-). To avoid lack of diversity at the beginning of the reads, at least four different plate indices were used for each sequencing run. Primers were ordered desalted as high-performance liquid chromatography purified. During lysis and each PCR, a mock control was added. The DNA amplification by the three PCRs was monitored by the run on a large 2% agarose gel. PCR3 products for each sample and replicate were pooled, purified with the Agencourt AMPure XP system (Beckman Coulter), diluted to 5 nM, and sequenced on a HiSeq system (Illumina) (SR-65bp) at the Institute Curie facility with 10% of PhiX spike-in.

## Barcode sequence analysis

Sequencing results were filtered, and barcodes were categorized in progenitor classes as in *Naik et al., 2013* and further explained on GitHub (https://github.com/PerieTeam/Eisele-et-al.-). In brief, sequencing results were analyzed using R-3.4.0 (R Development Core Team 2019; http://wwwR-project.org.), Excel, and GraphPad Prism version 8.0 for Mac (GraphPad Software, La Jolla, CA, https://www.graphpad.com). Reads were first filtered for perfect match to the input index- and common-sequences using XCALIBR (https://github.com/NKI-GCF/xcalibr) and filtered against the barcode reference list. Samples were then filtered for containing at least 5000 reads and normalized to $10^5$ per sample. Samples with a Pearson correlation between duplicates below 0.9 were discarded, and barcodes present in one of the two replicates were set to zero. Samples with under 10 barcodes

were filtered out, unless indicated in the figure legend. The mean of the replicates was used for further processing. When the mean percentage of barcodes shared between different sequencing runs was higher than within the same sequencing run for mice of a same transduction batch, reads below the read quartile of the mean percentage of barcodes shared between mice of a same transduction batch but sequenced on different sequencing runs were set to zero in order to equalize the barcode sharing between mice transplanted from a same transduction batch in different sequencing runs to the barcode sharing between mice within each sequencing run. After filtering, read counts of each barcode in the different cell lineages were normalized enabling categorization into classes of biased output toward the analyzed lineages using a threshold of 10% of barcode reads (other thresholds in *Figure 1—figure supplement 3c* and *Figure 7—figure supplement 1c*). Statistics on barcoding results were performed using a permutation test as in *Tak et al., 2021*. Significance of flow cytometry results was assessed using Student's *t*-test. Some mice were excluded from the analysis due to death before readout or due to a donor cell engraftment <5%, as well as the filtering out of mice for which one or more cell subset samples did not pass the barcode data filtering steps as detailed above.

## scRNAseq and analysis

scRNAseq was performed using the 10X Genomics platform on one pool of HSPCs isolated from eight mice, barcoded and culture with or without EPO for 16 hr in vitro as described above. Sequencing libraries were prepared using the Chromium Single Cell 3′ v2 kit and sequenced on a HiSeq system (Illumina) at the Institut Curie NGS facility. Data was analyzed using Cellranger v3 (10X Genomics), Seurat v3 (*Satija et al., 2015*), and customized scripts. Raw sequencing reads were processed using Cellranger. To obtain a reads/cell/gene count table, reads were mapped to the mouse GRCm38.84 reference genome. scRNAseq analysis was performed using Seurat (*Satija et al., 2015*). During filtering, Gm, Rik, and Rp genes were discarded as noninformative genes. Cells with less than 1000 genes per cell and with a high percentage of mitochondrial genes were removed from downstream analyses. Following our filtering procedures, the average UMI count per cell was 5157, with mitochondrial genes accounting for 5% of this. The average number of genes detected per cell was 2337. Cell cycle annotation using the cyclone method from the scran R package showed that 2938 cells were in G1 phase, 233 cells were in G2M phase, and 127 cells were in S phase. No batch effect was detected between the EPO and no-EPO group; therefore, no batch correction was applied. Data normalization was performed using the default Seurat approach, and differentially expressed genes were determined using a logistic regression in Seurat. Unsupervised clustering was performed on the significant variable genes using the 10 first PCA followed by the nonlinear dimensionality reduction technique UMAP (*McInnes et al., 2018*; *Figure 5—figure supplement 2*). Unsupervised Louvain clustering of the data was performed across a range of resolution parameters, and the resolution value that led to the most stable clustering profiles was chosen (*Blondel et al., 2008*; *Figure 5—figure supplement 2*). Annotation of the clusters was obtained by mapping published signatures using the *AddModuleScore* method of Seurat. The signatures are defined in the following publications: dHSC and MPP1 signatures were obtained from *Cabezas-Wallscheid et al., 2017*. The MolO LT-HSC signature was taken from *Wilson et al., 2015*, and the MPP2 and 4 signature was taken from *Pietras et al., 2015*. An Excel file listing the genes in these signatures is available on GitHub (https://github.com/PerieTeam/Eisele-et-al.-). To identify EPO responder cells in the EPO group, differential expression analysis was performed between the control and EPO groups (lists of DEGs are available at https://github.com/PerieTeam/Eisele-et-al.-). Subsequently, genes that were differentially expressed (adjusted p-value<0.05) between the EPO and control groups were transformed into an EPO response signature that when overlaid onto the UMAP-based visualization was enriched only in a subset of the EPO group cells. Briefly this signature was obtained by taking the background-corrected mean expression values of both the up- and downregulated genes per cell as implemented in the *AddModuleScore* method of Seurat. Within each cell, these two signature scores were used to create a composite EPO response score by subtracting the downregulated response from the upregulated response signature. Cells in the upper 90th percentile with regards to the expression of the EPO response signature were labeled EPO responders.

To perform supervised cell-type annotation, a reference map was generated from a published single-cell sequencing dataset of 44,802 C-Kit+ cells from *Dahlin et al., 2018*. Preprocessing was performed using a scanpy pipeline (*Wolf et al., 2019*). Data was then visualized using the nonlinear

nondimensionality reduction technique UMAP (*McInnes et al., 2018*). Annotation of the reference map was obtained by overlaying published signatures as above using the *AddModuleScore* method of Seurat and also known markers as *Flt3*, *slamf1*, and *Gata1* (*Figure 5—figure supplement 1b*). For the erythroid progenitors, these markers are *Gata1, Klf1, Epor, Gypa, Hba-a2, Hba-a1* (*Figure 5—figure supplement 1a*). Cells were mapped onto the reference map using a k-nearest-neighbors mapping approach. Briefly, for each cell in the query dataset, the nearest neighbors in the PCA space of the reference dataset were determined using the nn2 function of the RANN package, and the mean UMAP 1 and 2 coordinates of the 10 nearest neighbors were taken as the reference point for the new cell of interest. To benchmark our mapping approach, cells from an independent dataset of erythroid progenitors *Tusi et al., 2018* were used without additional preprocessing (*Figure 5—figure supplement 1a*).

## Data availability statement

Raw data are available at zenodo doi:10.5281/zenodo.5645045. All codes to filter and process raw data, as well as filtered data, are available at https://github.com/PerieTeam/Eisele-et-al.-. Contact author is leila.perie@curie.fr.

## Acknowledgements

We thank Dr. T Schumacher for discussion and lentiviral library production and Dr. K Duffy for advices on permutation testing. We thank the Institute Curie flow cytometry, next-generation sequencing, and animal facility. We thank Fahima Di Federico from the UMR168 BMBC facility for amplifying the barcode plasmid pool.

## Additional information

### Funding

| Funder | Grant reference number | Author |
|---|---|---|
| ATIP Avenir CNRS | | Leïla Perié |
| Labex Cell(n)scale | ANR-10-LABX-31 | Leïla Perié |
| Idex Paris-Science-Lettres | ANR-10-IDEX-0001-02 PSL | Leïla Perié |
| H2020 European Research Council | 758170-Microbar | Leïla Perié |
| H2020 Marie Skłodowska-Curie Actions | 666003 | Almut S Eisele |

The funders had no role in study design, data collection and interpretation, or the decision to submit the work for publication.

### Author contributions

Almut S Eisele, Conceptualization, Data curation, Formal analysis, Investigation, Methodology, Writing - original draft; Jason Cosgrove, Data curation, Formal analysis, Investigation, Visualization, Writing - original draft; Aurelie Magniez, Investigation, Writing - review and editing; Emilie Tubeuf, Sabrina Tenreira Bento, Cecile Conrad, Tamar Tak, Investigation; Fanny Cayrac, Investigation, Methodology; Anne-Marie Lyne, Formal analysis, Writing - review and editing; Jos Urbanus, Conceptualization, Methodology; Leïla Perié, Conceptualization, Formal analysis, Funding acquisition, Investigation, Methodology, Supervision, Writing - original draft

### Author ORCIDs

Almut S Eisele  http://orcid.org/0000-0001-8376-4089
Leïla Perié  http://orcid.org/0000-0003-0798-4498

### Ethics

All procedures were approved by the responsible national ethics committee (APAF-IS#10955-201708171446318 v1).

Decision letter and Author response
Decision letter https://doi.org/10.7554/eLife.66922.sa1
Author response https://doi.org/10.7554/eLife.66922.sa2

## Additional files

### Supplementary files

• Supplementary file 1. Permutation testing of changes in clonality after transplantation of erythropoietin (EPO)-exposed hematopoietic stem and progenitor cells (HSPCs). Same data as in *Figure 1—figure supplement 4* and *Figure 1—figure supplement 3*. HSPCs were cultured with EPO (1000 ng/ml) for 16 hr. Barcodes in the erythroid (E), myeloid (M), B-lymphoid (B) lineage, dendritic cell (DC), and HSPCs were analyzed 4 weeks after transplantation and categorized by bias using a 10% threshold. The output of MB and ME classified barcodes to the B, M, and E lineages was analyzed using a permutation test. By permutating the mice of the control and EPO groups, the random distribution of this output was generated and compared to the real output difference between the control and EPO group. A p-value was generated by permutation testing.

• Transparent reporting form

### Data availability

all data and scripts are available on the github of the Perie lab https://github.com/PerieTeam/Eisele-et-al.- (copy archived at swh:1:rev:ff1da6c9b3ec2b8e14e5921aeb2ac70fa2bcced0) .

The following dataset was generated:

| Author(s) | Year | Dataset title | Dataset URL | Database and Identifier |
|---|---|---|---|---|
| Eisele AS | 2020 | Erythropoietin directly remodels the clonal composition of murine hematopoietic multipotent progenitor cells | https://github.com/PerieTeam/Eisele-et-al.- | GitHub, GitHub |

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
