## [Editor Report]

This paper will be of broad interest to readers in the field of cytokine signaling, experimental hematology, and clinical hematology. Erythropoietin is one of the most widely used cytokines clinically, but the cells it exerts its effects on has been debated. This study has combined clonal lineage tracing and single-cell sequencing to understand the cell population that responds to erythropoietin and indicates that erythropoietin acts directly on multipotent progenitors to transiently modulate their output.

---

## [Decision Letter]

**Decision letter after peer review:**

Thank you for submitting your article "Erythropoietin directly remodels the clonal composition of murine hematopoietic multipotent progenitor cells" for consideration by *eLife*. Your article has been reviewed by 3 peer reviewers, one of whom is a member of our Board of Reviewing Editors, and the evaluation has been overseen by Utpal Banerjee as the Senior Editor. The following individual involved in review of your submission has agreed to reveal their identity: Carl R Walkley (Reviewer #3).

Essential revisions:

This study provides a detailed analysis of the effects of EPO exposure on HSPCs and the output from these cells after transplant. The analyses are well performed and technically sound. However, the conclusion needs validation by additional experiments and several points should be clarified or discussed more in detail.

1. Based on the scRNA-seq analysis, the authors conclude that "HSPCs directly reacting to EPO are most likely multipotent progenitor cells of the MPP1 and MPP2 subsets" (page 15). This should be directly tested by treating MPP1 and MPP2 with EPO and transplant, as they have done with LSK CD150+Flt3- cells.

2. How does EPO increase the relative contribution by ME and MB clones and reduce the contribution by MBE clones at 4 weeks after transplantation (Figure 1g) without affecting the overall reconstitution level at 4 weeks (Figure 1f) and the contribution by MBE clones at 16 weeks (Figure 7b)? Do the authors postulate that EPO transiently suppresses output by MBE clones (presumably LT-HSCs)? If so, this should be directly tested by incubating LT-HSCs (or the MBE clones) with EPO and transplant. Given that the overall reconstitution level did not change with EPO stimulation (Figure 1f), if some clones (e.g. ME and MB) increased their output, some cells must have reduced their output.

3. It was not clear why the doses of erythropoietin were chosen and why such a high dose was used in vivo. How and why was a dose of 16,000 IU/kg for in vivo injection used? This is a very high dose of Epo to use based on previous studies (for example Exp Hematol 31 (2003) 290-299). A better description of the choice of Epo doses and a consistency of nomenclature (all ng or IU not a mix of both) would be helpful for uniformity of the text.

Is there also a dose response that is apparent for MPP gene expression and fate changes to become apparent? By this I mean can effects be seen in these cell populations only above a threshold level of Epo and if this was the case how does this level of Epo compare to that required on erythroid progenitors? Do erythroid progenitors respond to a substantially lower Epo level? This would be important to understand in what contexts the MPP response to Epo can be enacted compared to the dominant response being from more mature and higher EpoR expressing populations. It would also be important to understand the pathological context in which MPP responses to Epo may be more apparent and for readers from different fields (clinical vs experimental hematology).

4. The cells were isolated, treated in vitro and then transplant was used. The behavior of cells that are transplanted compared to native hematopoiesis is distinct and has been demonstrated to yield a more differentiative output post-transplant where MPPs and more mature progenitors can contribute to steady state hematopoiesis. The authors should include discussion of this with respect to their results and those available from either analysis post Epo injection in native hematopoiesis or from other models with elevated Epo levels (such as Epo Tg mice such as in PMID: 29754961).

The duration of exposure of the cell to Epo, the duration of signaling from a singular treatment/dose of Epo vs sustained or long acting Epo and the impact of sustained changes in lineage output (from systemic elevations of Epo for weeks/months) are worth considering when discussing how these results of the present study fit within the field. The data in the present work support an effect of Epo at the MPP stage of the hematopoietic hierarchy.

5. Please move the colored plot from Supp Figure 3E to Figure 1E – this is a very good figure to assist the reader understanding the data presentation and where the different cell populations reside.

[Editors' note: further revisions were suggested prior to acceptance, as described below.]

Thank you for resubmitting your work entitled "Erythropoietin directly remodels the clonal composition of murine hematopoietic multipotent progenitor cells" for further consideration by *eLife*. Your revised article has been evaluated by Utpal Banerjee (Senior Editor) and a Reviewing Editor.

The manuscript has been much improved but there are two remaining issues that need to be clarified before acceptance, as follows:

1. The authors write "a slight downregulation of c-kit and Flt3 (line 120)". However, Figure 1 S1f appears to be showing increased Flt3 expression (or an appearance of Flt3+ fraction) after EPO.

2. The authors write "with the majority of cells in all lineages produced by balanced HSPCs (Figure 8a-b) (line 316)". While the majority of the cells are indeed from MBE, there seems to be more erythroid and myeloid contribution by ME cells after EPO even at 4 months. Are the differences between the ME contribution to these lineages with or without EPO statistically significant?

Please clarify the above two points in your revised article.

*Reviewer #1 (Recommendations for the authors):*

The authors performed additional experiments such as direct treatment of MPP2 with EPO followed by transplantation, which showed the EPO treated MPP2 are really enriched in myeloid-erythroid biased cells. Although they avoided doing some of the experiments due to the technical limitations, they responded to most of the key questions and redrafted the manuscript precisely.

*Reviewer #2 (Recommendations for the authors):*

The authors provide new data showing that MPP2 are capable of responding to EPO in vitro and become more ME-biased, which was the main concern for this reviewer.

Some clarifications are needed as follows.

1. The authors write "a slight downregulation of c-kit and Flt3 (line 120)". However, Figure 1 S1f appears to be showing increased Flt3 expression (or an appearance of Flt3+ fraction) after EPO.

2. The authors write "with the majority of cells in all lineages produced by balanced HSPCs (Figure 8a-b) (line 316)". While the majority of the cells are indeed from MBE, there seems to be more erythroid and myeloid contribution by ME cells after EPO even at 4 months. Are the differences between the ME contribution to these lineages with or without EPO statistically significant?

---

## [Author Response]

Essential revisions:This study provides a detailed analysis of the effects of EPO exposure on HSPCs and the output from these cells after transplant. The analyses are well performed and technically sound. However, the conclusion needs validation by additional experiments and several points should be clarified or discussed more in detail.

We thank the reviewers for careful reading of manuscript and constructive suggestions to improve our manuscript. We have performed required additional experiments to validate our conclusion and have clarified the text to answer the points that were raised. You will find a detailed answer to each point below.

1. Based on the scRNA-seq analysis, the authors conclude that "HSPCs directly reacting to EPO are most likely multipotent progenitor cells of the MPP1 and MPP2 subsets" (page 15). This should be directly tested by treating MPP1 and MPP2 with EPO and transplant, as they have done with LSK CD150+Flt3- cells.

We have now directly tested our scRNAseq conclusion focusing on the MPP2 subset, which we predicted are the major contributors to the EPO response. Specifically, barcoded MPP2 (LSK CD150+FLt3-CD48+), together with the unbarcoded remainder of the HSPC compartment (LSK CD150+FLt3-CD48-) were transplanted into irradiated recipients and were harvested for barcode analysis 1 month later.

The results corroborate our scRNAseq conclusions showing that relative to control the EPO treated MPP2 are enriched in myeloid-erythroid biased cells. These new results are now in Figure 7, Figure 7 —figure supplement 1, and lines 476-489 of the main text.

2. How does EPO increase the relative contribution by ME and MB clones and reduce the contribution by MBE clones at 4 weeks after transplantation (Figure 1g) without affecting the overall reconstitution level at 4 weeks (Figure 1f) and the contribution by MBE clones at 16 weeks (Figure 7b)? Do the authors postulate that EPO transiently suppresses output by MBE clones (presumably LT-HSCs)? If so, this should be directly tested by incubating LT-HSCs (or the MBE clones) with EPO and transplant. Given that the overall reconstitution level did not change with EPO stimulation (Figure 1f), if some clones (e.g. ME and MB) increased their output, some cells must have reduced their output.

From the data we have generated, we observe reduced MBE clonal output at 4 weeks post EPO treatment relative to control. Overall reconstitution levels are maintained at 4 weeks because the increased ME and MB contribution is balanced by the reduced contribution of the MBE clones. At 16 weeks the output of MBE is comparable in the EPO and no EPO group, also the overall number of cells produced is increased compared to 4 weeks. However, from our data we don’t wish to conclude that EPO transiently suppress MBE clones and transiently increase ME and MB from the same individual HSPC over time. To draw this conclusion, one would need to have longitudinal barcoding data within the same animal to see if it is the same barcodes that contribute over time.

The experiment proposed by the reviewers is to test the conclusion that EPO stimulation leads to a direct suppressive effect on MBE clonal dynamics. Such an experiment would require sorting MBE, ME or MB clones. Currently we have no surface markers to purify these functional subgroups and can at best enrich for EPO-responding cells based on the scRNAseq data as we did for MPP2. We believe the question asked by the reviewers is interesting and we have now added further discussion on this point in the Discussion section (line 542-545).

3. It was not clear why the doses of erythropoietin were chosen and why such a high dose was used in vivo. How and why was a dose of 16,000 IU/kg for in vivo injection used? This is a very high dose of Epo to use based on previous studies (for example Exp Hematol 31 (2003) 290-299). A better description of the choice of Epo doses and a consistency of nomenclature (all ng or IU not a mix of both) would be helpful for uniformity of the text.

Regarding the concentration of EPO, we have changed the indications of EPO concentration into ng/ml throughout as asked by the reviewer (120 IU are 1 ug). The EPO doses were chosen based on previous studies testing the effect of EPO on HSPCs. We have summarized the concentration used in our study and these studies in Author response table 1. The EPO concentrations we used are in similar range to these studies, in vitro and in vivo. All these doses are above physiological level (around 100-200 pg/ml in blood EPO in humans and mice) but close to the dose in anemia or arterial hypoxemia where EPO levels can rise 1000 fold ((Kaushansky, 2006) (Elliott & Sinclair, 2012)). In the clinics, the American society of hematology recommends 1.25 ug/kg twice a week for a minimum of 4 weeks with consideration for dose escalation to 2.5 ug/kg for an additional 4-8 weeks.

**Author response table 1. sa2table1:** 

**Reference**	**in vitro EPO dose**	**in vivo EPO dose (route, duration )**
Eisele et al.,	160-1000ng/ml	133 ug/kg (1 single injection)
Grover et al., 2014	200ng/ml	1µg/ml (plasma level, continous)
Giladi et al., 2018		83 ug/kg (injection/day, for 2 days)
Shiozawa et al., 2010		50ug/kg (3 injection/week, for 28 days)
Tusi et al., 2018		33 ug/kg (injection/day, for 2 days)

Is there also a dose response that is apparent for MPP gene expression and fate changes to become apparent? By this I mean can effects be seen in these cell populations only above a threshold level of Epo and if this was the case how does this level of Epo compare to that required on erythroid progenitors? Do erythroid progenitors respond to a substantially lower Epo level? This would be important to understand in what contexts the MPP response to Epo can be enacted compared to the dominant response being from more mature and higher EpoR expressing populations. It would also be important to understand the pathological context in which MPP responses to Epo may be more apparent and for readers from different fields (clinical vs experimental hematology).

Concerning the question about dose response and effect on fate and gene expression, this is exactly the reason for which we had performed the dose response in the original submission (Figure 2). At EPO dose ranging from 160 ng/ml to 1000 ng/ml, we always observed a higher proportion of myeloid-erythroid biased HSPC compared to control. We do not observe any threshold effect at these doses.

Comparing the dose at which HSPC and erythroid progenitors respond to EPO is very interesting but is out of scope for this study. Indeed, here we wish to answer the question whether HSPC can directly respond to EPO, which was still a debate in the field, and characterize this response at the single cell level.

Please also consider that the functional profiling of EPO responses in MPP vs erythroid progenitors across a range of EPO concentrations is a study in and of itself. Firstly, committed erythroid progenitors have never been used in lentiviral barcoding and so establishing this method in a new cell type is non-trivial. Secondly, HSPCs are a rare and functionally heterogeneous population of cells that require complicated transplantation assays using large cohorts of experimental animals to draw meaningful conclusions.

We have now added some sentences related to the response of HSPC versus erythroid progenitors to EPO in the Discussion section (lines 555-557).

4. The cells were isolated, treated in vitro and then transplant was used. The behavior of cells that are transplanted compared to native hematopoiesis is distinct and has been demonstrated to yield a more differentiative output post-transplant where MPPs and more mature progenitors can contribute to steady state hematopoiesis. The authors should include discussion of this with respect to their results and those available from either analysis post Epo injection in native hematopoiesis or from other models with elevated Epo levels (such as Epo Tg mice such as in PMID: 29754961).The duration of exposure of the cell to Epo, the duration of signaling from a singular treatment/dose of Epo vs sustained or long acting Epo and the impact of sustained changes in lineage output (from systemic elevations of Epo for weeks/months) are worth considering when discussing how these results of the present study fit within the field. The data in the present work support an effect of Epo at the MPP stage of the hematopoietic hierarchy.

We added several sentences in the Discussion section (lines 530-551) to discuss:

1. The difference between transplantation and native hematopoiesis.

2. The duration of exposure of the cell to EPO, the duration of signaling from a singular treatment/dose of EPO vs sustained or long acting EPO and the impact of sustained changes in lineage output (from systemic elevations of EPO for weeks/months).

5. Please move the colored plot from Supp Figure 3E to Figure 1E – this is a very good figure to assist the reader understanding the data presentation and where the different cell populations reside.

We moved Supplementary Figure 3b to the main figure 1, as asked for by the reviewers.

[Editors' note: further revisions were suggested prior to acceptance, as described below.]

The manuscript has been much improved but there are two remaining issues that need to be clarified before acceptance, as follows:1. The authors write "a slight downregulation of c-kit and Flt3 (line 120)". However, Figure 1 S1f appears to be showing increased Flt3 expression (or an appearance of Flt3+ fraction) after EPO.

We thank the reviewer for highlighting this point which was actually a mistake from our side.

Flt3 is indeed a bit upregulated following in vitro culture. The upregulation is slightly lower in EPO treated compare to control but the difference is not consistent across experiment so we are not commenting on this point. We have modified the text as follows:

“Note that HSPCs kept their sorting phenotype after ex vivo culture albeit a slight downregulation of C-Kit^44^ and upregulation of Flt3 (Figure 1 —figure supplement 1f).”

2. The authors write "with the majority of cells in all lineages produced by balanced HSPCs (Figure 8a-b) (line 316)". While the majority of the cells are indeed from MBE, there seems to be more erythroid and myeloid contribution by ME cells after EPO even at 4 months. Are the differences between the ME contribution to these lineages with or without EPO statistically significant?

We thank the reviewer for this question which revealed an important point we had missed in our first analysis. After doing the permutation testing, it appears that the ME contribution to myeloid and erythroid cells is still statistically significant at 4 months. It is an important point and we have changed our phrasing in the main text. The general conclusion remains unchanged however as the majority of the cells come from MBE clones by the 4 month timepoint, indicating that the EPO effect is fading away. The text has been modified and now reads as follows:

“While ME-biased HSPCs still have a significantly higher contribution of to the M and E lineage (Figure 8a-b, Table 1), the majority of cells in all lineages were produced by balanced HSPCs (Figure 8a-b), implying that the effect of direct EPO-ex